# Inhibition of ErbB kinase signalling promotes resolution of neutrophilic inflammation

Atiqur Rahman[1,2], Katherine M Henry[1,3], Kimberly D Herman[1,3], Alfred AR Thompson[1], Hannah M Isles[1,3], Claudia Tulotta[1,3], David Sammut[1], Julien JY Rougeot[4‡], Nika Khoshaein[1], Abigail E Reese[1], Kathryn Higgins[1], Caroline Tabor[3], Ian Sabroe[1], William J Zuercher[5], Caroline O Savage[6‡], Annemarie H Meijer[4], Moira KB Whyte[7], David H Dockrell[1,7], Stephen A Renshaw[1,3†], Lynne R Prince[1†*]

[1]Department of Infection, Immunity and Cardiovascular Disease, University of Sheffield, Sheffield, United Kingdom; [2]Department of Biochemistry and Molecular Biology, Faculty of Biological Sciences, University of Dhaka, Dhaka, Bangladesh; [3]The Bateson Centre, University of Sheffield, Sheffield, United Kingdom; [4]Institute of Biology, Leiden University, Leiden, Netherlands; [5]SGC-UNC, Division of Chemical Biology and Medicinal Chemistry, UNC Eshelman School of Pharmacy, University of North Carolina at Chapel Hill, Chapel Hill, United States; [6]Immuno-Inflammation Therapy Area Unit, GlaxoSmithKline Research and Development Ltd, Stevenage, United Kingdom; [7]MRC Centre for Inflammation Research, University of Edinburgh, Edinburgh, United Kingdom

*For correspondence:
L.r.prince@sheffield.ac.uk

†These authors contributed equally to this work

Present address: ‡Institute of Immunology and Immunotherapy, University of Birmingham, Birmingham, United Kingdom

**Abstract** Neutrophilic inflammation with prolonged neutrophil survival is common to many inflammatory conditions, including chronic obstructive pulmonary disease (COPD). There are few specific therapies that reverse neutrophilic inflammation, but uncovering mechanisms regulating neutrophil survival is likely to identify novel therapeutic targets. Screening of 367 kinase inhibitors in human neutrophils and a zebrafish tail fin injury model identified ErbBs as common targets of compounds that accelerated inflammation resolution. The ErbB inhibitors gefitinib, CP-724714, erbstatin and tyrphostin AG825 significantly accelerated apoptosis of human neutrophils, including neutrophils from people with COPD. Neutrophil apoptosis was also increased in Tyrphostin AG825 treated-zebrafish in vivo. Tyrphostin AG825 decreased peritoneal inflammation in zymosan-treated mice, and increased lung neutrophil apoptosis and macrophage efferocytosis in a murine acute lung injury model. Tyrphostin AG825 and knockdown of *egfra* and *erbb2* by CRISPR/Cas9 reduced inflammation in zebrafish. Our work shows that inhibitors of ErbB kinases have therapeutic potential in neutrophilic inflammatory disease.
DOI: https://doi.org/10.7554/eLife.50990.001

## Introduction

Neutrophilic inflammation is central to chronic inflammatory diseases such as rheumatoid arthritis and chronic obstructive pulmonary disease (COPD), which impose an increasing social and economic burden on our aging population. Treatment of COPD by next-generation combination therapy with inhaled corticosteroids and newer bronchodilators are viewed as maintenance pharmacotherapies but they do not specifically target cellular inflammation. The anti-inflammatory phosphodiesterase-4 inhibitor, roflumilast, targets systemic inflammation associated with COPD and reduces moderate to

**eLife digest** Chronic obstructive pulmonary disease (or COPD) is a serious condition that causes the lungs to become inflamed for long periods of time, leading to permanent damage of the airways.

Immune cells known as neutrophils promote inflammation after an injury, or during an infection, to aid the healing process. However, if they are active for too long, they may also cause tissue damage and drive inflammatory diseases including COPD. To limit damage to the body, neutrophils usually have a very short lifespan and die by a regulated process known as apoptosis. Finding ways to stimulate apoptosis in neutrophils may be key to developing better treatments for inflammatory diseases.

Cells contain many enzymes known as kinases that control apoptosis and other cell processes. Drugs that inhibit specific kinases are effective treatments for some types of cancer and other conditions, and new kinase-inhibiting drugs are currently being developed. However, it remains unclear which kinases regulate apoptosis in neutrophils or which kinase-inhibiting drugs may have the potential to treat COPD and other inflammatory diseases.

To address these questions, Rahman et al. tested over 350 kinase-inhibiting drugs to identify ones that promote apoptosis in neutrophils. The experiments showed that human neutrophils treated with drugs that inhibit the ErbB family of kinases died by apoptosis more quickly than untreated neutrophils. Next, Rahman et al. used zebrafish with injured tail fins as models to study inflammation. Zebrafish treated with one of these drugs – known as Tyrphostin AG825 – had lower levels of inflammation and their neutrophils underwent apoptosis more frequently than untreated zebrafish. Since drugs can have off-target effects, Rahman et al. went on to show using gene-editing technology that reducing the activity of two genes that encode ErbB kinases in zebrafish also decreased the levels of inflammation in the fish.

Further experiments used mice that develop inflammation in the lungs similar to COPD in humans. As expected, neutrophils in the lungs of mice treated with Tyrphostin AG825 underwent apoptosis more frequently than those in untreated mice. These dead neutrophils were effectively cleared by other immune cells called macrophages, which also helps limit damage caused by neutrophils.

Together, these findings show that Tyrphostin AG825 and other drugs that inhibit ErbB kinases help to reduce inflammation by promoting the death of neutrophils. Since several of these drugs are already used to treat human cancers, it may be possible in the future to repurpose them for use in people with COPD and other long-term inflammatory diseases. Determining whether this is possible is an aim for future studies.

DOI: https://doi.org/10.7554/eLife.50990.002

---

severe exacerbations in severe disease, possibly via effects on eosinophils (*Martinez et al., 2018*; *Rabe et al., 2018*). In these diseases, clearance of neutrophils by apoptosis is dysregulated particularly during exacerbations (*Pletz et al., 2004*; *Sapey et al., 2011*), but to date it has not been possible to therapeutically modify this, indeed corticosteroids can supress neutrophil apoptosis and hence perpetuate inflammation (*Liles et al., 1995*). Recognising the urgent need for new therapies, we interrogated neutrophil inflammation and survival pathways using an unbiased approach focusing on potentially druggable kinases. Neutrophil persistence in tissues, caused by a delay in apoptosis, can result in a destructive cellular phenotype, whereby neutrophils have greater potential to expel histotoxic factors such as proteases and oxidative molecules onto surrounding tissue. This can occur either actively (by degranulation) or passively (by secondary necrosis). In COPD, among other diseases, delayed apoptosis is considered to be a key part of the pathogenesis, occurring either as a result of pro-survival factors that are present in the lung microenvironment or an innate apoptosis defect (*Brown et al., 2009*; *Haslett, 1999*; *Pletz et al., 2004*; *Zhang et al., 2012*). Despite this mechanistic understanding, there are no effective treatment strategies in clinical use to specifically reverse this cellular mechanism.

Accelerating neutrophil apoptosis has been shown to promote the resolution of inflammation in multiple experimental models (*Burgon et al., 2014*; *Chello et al., 2007*; *Ren et al., 2008*;

*Rossi et al., 2006*). A number of studies highlight the importance of protein kinases in regulating neutrophil apoptosis (*Burgon et al., 2014*; *Rossi et al., 2006*; *Webb et al., 2000*) and therefore reveal potential therapeutically targetable pathways for inflammatory disease. A growing class of clinically-exploited small molecule kinase inhibitors are being intensively developed (*Wu et al., 2015*), making this a timely investigation. Using parallel unbiased screening approaches in vitro and in vivo, we here identify inhibitors of the ErbB family of receptor tyrosine kinases (RTKs) as potential therapeutic drivers of inflammation resolution. The ErbB family consist of four RTKs with structural homology to the human epidermal growth factor receptor (EGFR/ErbB1/Her-1). In an in vivo zebra-fish model of inflammation, we show that inhibition of ErbBs, pharmacologically and genetically, reduced the number of neutrophils at the site of injury. Furthermore, ErbB inhibitors reduced inflammation in a murine peritonitis model and promoted neutrophil apoptosis and clearance by macrophages in the mouse lung. This study reveals an opportunity for the use of ErbB inhibitors as a treatment for chronic neutrophilic inflammatory disease.

## Results

### Identifying kinases regulating the resolution of neutrophilic inflammation in vivo

Using a well-characterised transgenic zebrafish inflammation model (*Henry et al., 2013*; *Renshaw et al., 2006*), we adopted a chemical genetics approach, which has great potential for accelerated drug discovery (*Jones and Bunnage, 2017*). We initiated inflammation by controlled tissue injury of the zebrafish tail fin and screened a library of kinase inhibitors in order to establish which kinases could be exploited to enhance inflammation resolution in vivo (*Figure 1—figure supplement 1A*). We quantified the ability of a library of 367 publicly available kinase inhibitors (PKIS) (*Elkins et al., 2016*) to reduce neutrophil number at the site of injury during the resolution phase of inflammation. The screen identified 16 hit compounds which reduced neutrophil number at the site of injury in the zebrafish model (*Figure 1A*). For each compound the degree of kinase inhibition had been established (*Elkins et al., 2016*) (*Figure 1A*). A number of kinases were inhibited by the 16 compounds, with Abelson murine leukaemia viral homolog 1 (ABL1), Platelet-derived growth factor receptor (PDGFR) α, PDGFRβ, p38α and ErbB4 being the top five most frequently targeted kinases overall. In addition to frequency of target, we also interrogated selectivity of compound. The most selective compounds, that is those that strongly inhibited individual kinases or kinase families, targeted the kinases YES, ABL1, p38 and the ErbB family. Apoptosis is an important mechanism contributing to inflammation resolution; we therefore sought to identify kinases common to both inflammation resolution and neutrophil apoptosis pathways.

### Identifying kinases regulating neutrophil apoptosis in vitro

Circulating neutrophils have a short half-life in vivo (*Summers et al., 2010*) and undergo spontaneous apoptosis in the absence of growth factors in vitro. We re-screened PKIS library compounds in a human neutrophil apoptosis assay for their ability to accelerate apoptosis (*Figure 1—figure supplement 1B*). PKIS compounds were screened at 62 µM in order to maximise the chance of identifying 'hits' and resulted in 62 compounds that accelerated neutrophil apoptosis $\geq 2$ fold compared to DMSO control (*Figure 1B* and *Supplementary file 1*). Secondary screening of top 38 compounds (chosen from the 62 hits based on greatest selectivity for kinase targets) was carried out at 10 µM in order to reduce false positives. This yielded 11 compounds that accelerated neutrophil apoptosis $\geq 2$ fold over control (as indicated by dashed green line, *Figure 1C*). Representative flow cytometry dot plots illustrating Annexin-V and ToPro-3 profiles for these hit compounds are shown in *Figure 1—figure supplement 2*. Kinases targeted by these compounds included DYRK1B, KIT, EGFR, ErbB2 and ErbB4, PDGFR, CDK6 and p38 (*Figure 1C*, inset). The identification of known regulators of neutrophil survival (p38, PI3K) was encouraging support for the screen design and execution. We found that members of the ErbB family of RTKs were the next most frequently inhibited kinase family, being targeted by three highly selective compounds out of the 11 hits (*Figure 1C*, inset). Since inhibitors of the ErbB family were common hits in both zebrafish and human screens, we hypothesised that targeting ErbBs may be a potential strategy to reduce inflammation.

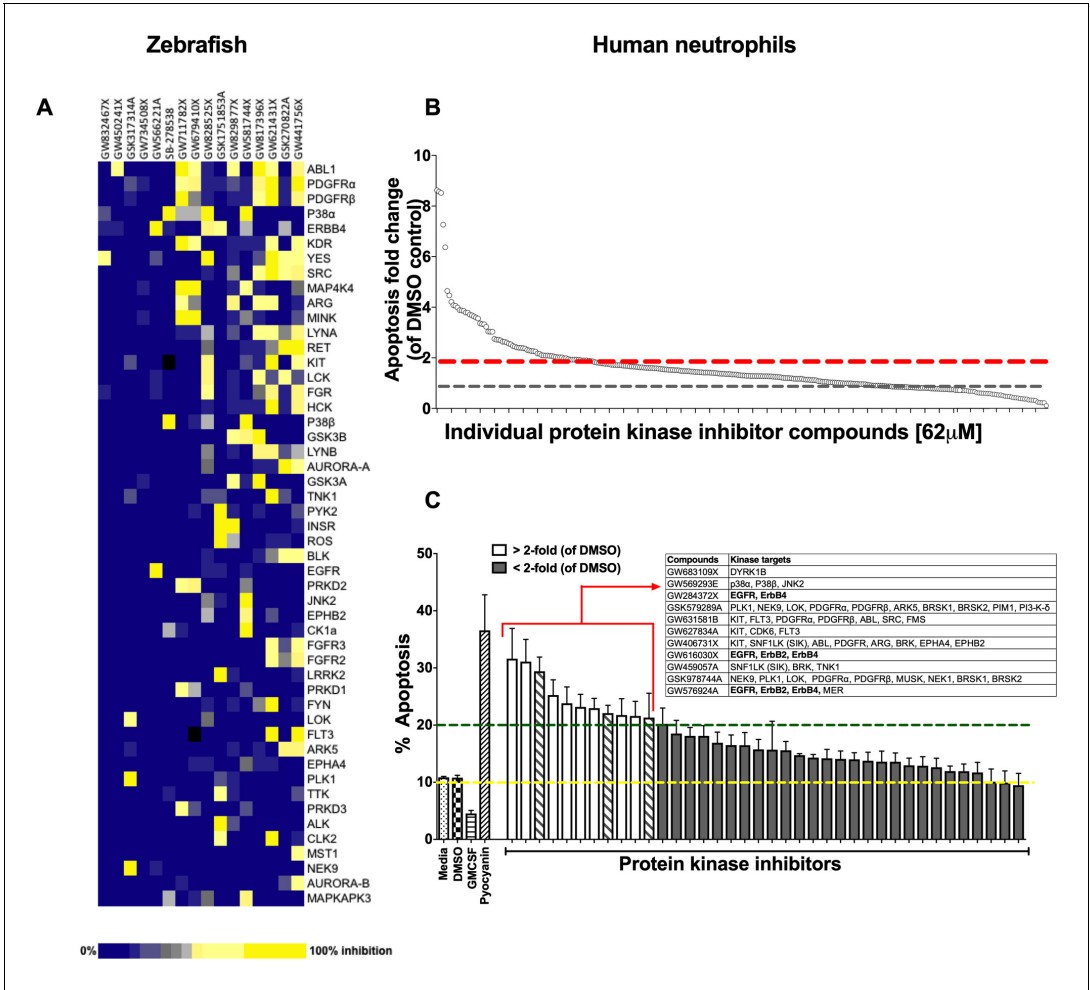

**Figure 1.** A protein kinase inhibitor compound library screen identifies compounds that promote the resolution of inflammation in vivo and neutrophil apoptosis in vitro. (**A**) *mpx:GFP* zebrafish larvae (three dpf) that had undergone tail fin transection resulting in an inflammatory response at six hpi were incubated with individual PKIS compounds [25 μM] three larvae/well for a further 6 hr. Wells were imaged and manually scored between 0–3 on the basis of GFP at the injury site in the larvae. 'Hit' compounds scored ≥1.5 (n = 2, three larvae per compound per experiment). Publicly available kinase profiling information was generated previously by *Elkins et al. (2016)* and kinase inhibition of each compound [1 μM] is shown as a gradient of blue to yellow. Hit compounds were ranked horizontally (left to right) from the most to least selective. Kinases (listed on the right) were vertically ranked from top to bottom from the most to least commonly targeted by inhibitors in PKIS. (**B**) PKIS compounds were incubated with primary human neutrophils for 6 hr. The entire library, at [62 μM], was screened on five separate days using five individual donors. Apoptosis was assessed by Annexin-V/TO-PRO-3 staining by flow cytometry and the percentage apoptosis calculated as Annexin-V single plus Annexin-V/TO-PRO-3 dual positive events. Data are expressed as fold change over DMSO control and each circle represents a single compound. Sixty two compounds accelerated apoptosis ≥2 fold as identified by red dotted line (n = 1). Grey dotted line represents level of apoptosis in DMSO control (i.e. no change). (**C**) Of the 62 compounds identified above, 38 of the most specific inhibitors were incubated with neutrophils at [10 μM] for 6 hr and apoptosis measured as above. Controls included media, DMSO, GMCSF [50 u/mL] and pyocyanin [50 μM]. Eleven compounds (white bars) accelerated apoptosis ≥2 fold over DMSO control (as identified by dotted line). Kinases targeted by the 11 compounds are shown in the inset table. Hatched bars represent data points in which ErbB inhibitors were used. Data are expressed as percentage apoptosis ± SEM, n = 3 neutrophil donors.

DOI: https://doi.org/10.7554/eLife.50990.003

The following figure supplements are available for figure 1:

**Figure supplement 1.** Schematics showing PKIS screen design.
DOI: https://doi.org/10.7554/eLife.50990.004

**Figure supplement 2.** Flow cytometry dot plots for screen validation.
DOI: https://doi.org/10.7554/eLife.50990.005

## ErbB inhibitors accelerate neutrophil apoptosis

To address a role for ErbB antagonists in regulating neutrophil apoptosis we tested a range of clinical and non-clinical ErbB-targeting compounds. We show that among inhibitors of ErbBs that are in clinical use, the EGFR inhibitor, gefitinib, is the most effective in promoting neutrophil apoptosis, reaching significance at 50 µM (*Figure 2A*). The ErbB2-selective inhibitor, CP-724714 (*Jani et al., 2007*) also promoted neutrophil apoptosis in a dose-dependent manner (*Figure 2B*) as did Erbstatin

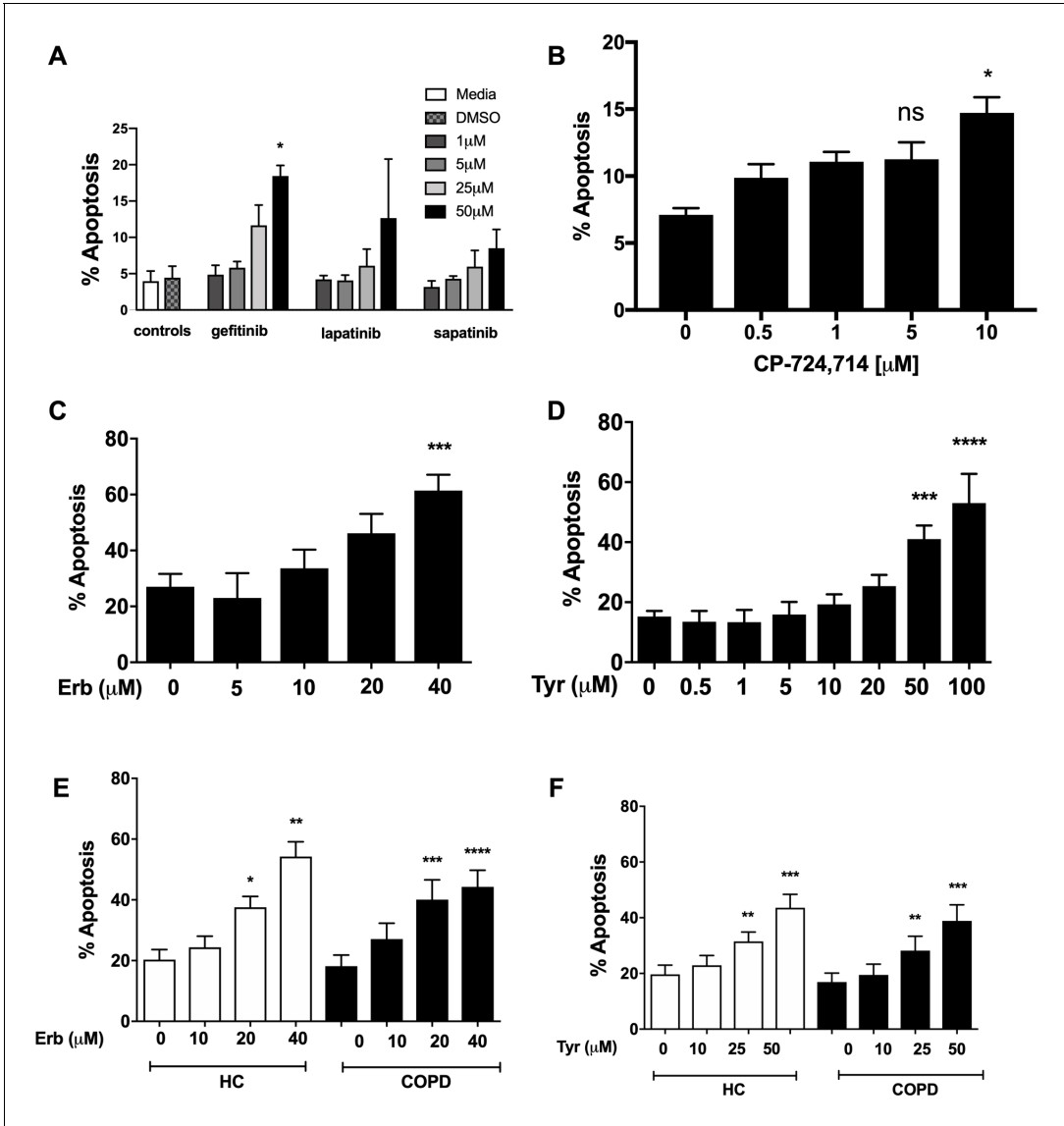

**Figure 2.** Inhibition of EGFR and ErbB2 drives apoptosis of neutrophils isolated from COPD patients and healthy subjects. Neutrophils were incubated with media or a concentration range of gefitinib (**A**), lapatinib (**A**), sapatinib (**A**), CP-724714 (**B**), erbstatin (Erb, **C**) or tyrphostin AG825 (Tyr, **D**) for 6 hr. Stars represent significant difference compared to DMSO control (indicated by '0' in B-D). Neutrophils from COPD patients (open bars) and age-matched healthy control subjects (black bars) were incubated with DMSO or a concentration range of erbstatin (**E**) or tyrphostin AG825 (**F**) for 6 hr. Apoptosis was assessed by light microscopy. The data are expressed as mean percentage apoptosis ± SEM from 3 (**B, D**), 4 (**A,C**), 10 (**E,F** COPD), or 7 (**E,F** HC) independent experiments using different neutrophil donors. Statistical significances between control and inhibitor was calculated by one-way ANOVA with Dunnett's post-test, indicated as *p<0.05, **p<0.01, ***p<0.001, ****p<0.0001.

DOI: https://doi.org/10.7554/eLife.50990.006

The following figure supplement is available for figure 2:

**Figure supplement 1.** Erbstatin and tyrphostin AG825 induces caspase-dependent neutrophil apoptosis.

DOI: https://doi.org/10.7554/eLife.50990.007

and tyrphostin AG825, selective for EGFR and ErbB2 respectively (*Osherov et al., 1993*; *Umezawa and Imoto, 1991*) (*Figure 2C–D*). Since caspase-dependent apoptosis is an anti-inflammatory and pro-resolution form of cell death, engagement of the apoptosis programme was verified biochemically by measuring phosphatidylserine (PS) exposure by Annexin-V staining (*Figure 2—figure supplement 1A–C*). Furthermore, the pan-caspase inhibitor Q-VD-OPh (*Wardle et al., 2011*) completely abrogated Erbstatin and tyrphostin AG825-driven neutrophil apoptosis, confirming the caspase dependence of inhibitor mediated cell death (*Figure 2—figure supplement 1D–E*).

COPD is a chronic inflammatory disease associated with functionally defective circulating neutrophils, including a resistance to undergoing apoptosis during exacerbations (*Pletz et al., 2004*; *Sapey et al., 2011*). To show ErbB inhibition is effective in driving apoptosis in subjects with systemic inflammation, we isolated neutrophils from the blood of patients with COPD and age-matched healthy control subjects. Erbstatin and tyrphostin AG825 significantly increased apoptosis of neutrophils from both COPD patients and healthy control subjects in a dose dependent manner at both 6 hr (*Figure 2E–F*) and 20 hr (data not shown).

ErbB inhibition overcomes neutrophil survival stimuli. Neutrophils are exposed to multiple pro-survival stimuli at sites of inflammation, which could undermine the therapeutic potential of anti-inflammatory drugs. Factors that raise intracellular cAMP concentration ([cAMP]$_i$) are present during inflammation, and elevated [cAMP]$_i$ is known to prolong neutrophil survival via activation of cAMP-dependent protein kinases (*Krakstad et al., 2004*; *Vaughan et al., 2007*). We show that neutrophil apoptosis was reduced by the cAMP analogue and site selective activator of PKA, $N^6$-monobutyryl-cAMP ($N^6$-MB-cAMP), and that this was reversed by Erbstatin analog (*Figure 3A*) and Tyrphostin AG825 (*Figure 3B*). Similar effects were observed in neutrophils from patients with COPD (*Figure 3C*). GMCSF is a key neutrophil chemoattractant and pro-survival factor, and is closely associated with the severity of inflammation in disease (*Klein et al., 2000*; *Wicks and Roberts, 2016*). We show that erbstatin and tyrphostin AG825 prevent GMCSF-mediated survival in COPD and age-matched healthy control neutrophils (*Figure 3D–E*). GMCSF is known to promote neutrophil survival via the phosphatidylinositol 3-kinase (PI3K)/AKT pathway, ultimately leading to the stabilisation of the anti-apoptotic Bcl-2 family member, Mcl-1 (*Derouet et al., 2004*; *Klein et al., 2000*). To investigate potential mechanisms underpinning the ability of tyrphostin AG825 to prevent GMCSF-mediated survival, we assessed AKT-phosphorylation as a measure of PI3K activation and found that tyrphostin AG825 reduced GMCSF-induced AKT phosphorylation after 15 and 30 min of treatment becoming statistically significant at 15 mins (*Figure 3F*). Tyrphostin AG825 accelerated the spontaneous downregulation of Mcl-1 and also prevented GMCSF-induced stabilisation of Mcl-1 (*Figure 3G*). These data show ErbB inhibition engages neutrophil apoptosis even in the presence of inflammatory stimuli and therefore has the potential to drive apoptosis at inflammatory sites.

## Kinase microarray profiling reveals ErbB2 is phosphorylated by neutrophil survival stimuli

To explore whether ErbB family members are phosphorylated in response to survival stimuli we studied the activated kinome in human neutrophils stimulated with $N^6$-MB-cAMP (*Vaughan et al., 2007*). A Kinex antibody microarray was performed to detect the phosphorylation of over 400 kinases and kinase-associated proteins and this data set was interrogated to seek evidence of activation of ErbB by $N^6$-MB-cAMP. Of the phospho-specific antibodies, 17 yielded an increase over baseline control of ≥1.5 at 30 min and 8 at 60 min (*Table 1*). Among these targets, ErbB2 phosphorylation was detected at 30 min (1.94 > control) and 60 min (1.53 > control, *Table 1*). This suggests that ErbB is part of the neutrophil signalling response to survival stimuli. In support of this, we detected the presence of ErbB2 mRNA in human neutrophils by RT-PCR (*Figure 4A*) and a 60kD protein (lower molecular weight ErbB family products are well-documented (*Jackson et al., 2013*; *Guillaudeau et al., 2012*; *Siegel, 1999*; *Ward et al., 2013*) which was upregulated by GMCSF and dbcAMP (*Figure 4B*). ErbB3 was also detected in human neutrophils by ELISA (*Figure 4C*), at levels similar to those observed in other tissues in literature (*Buta et al., 2016*). We found ErbB3 expression was not regulated by growth factors, which may in part be due to regulation being primarily at the post-translational level.

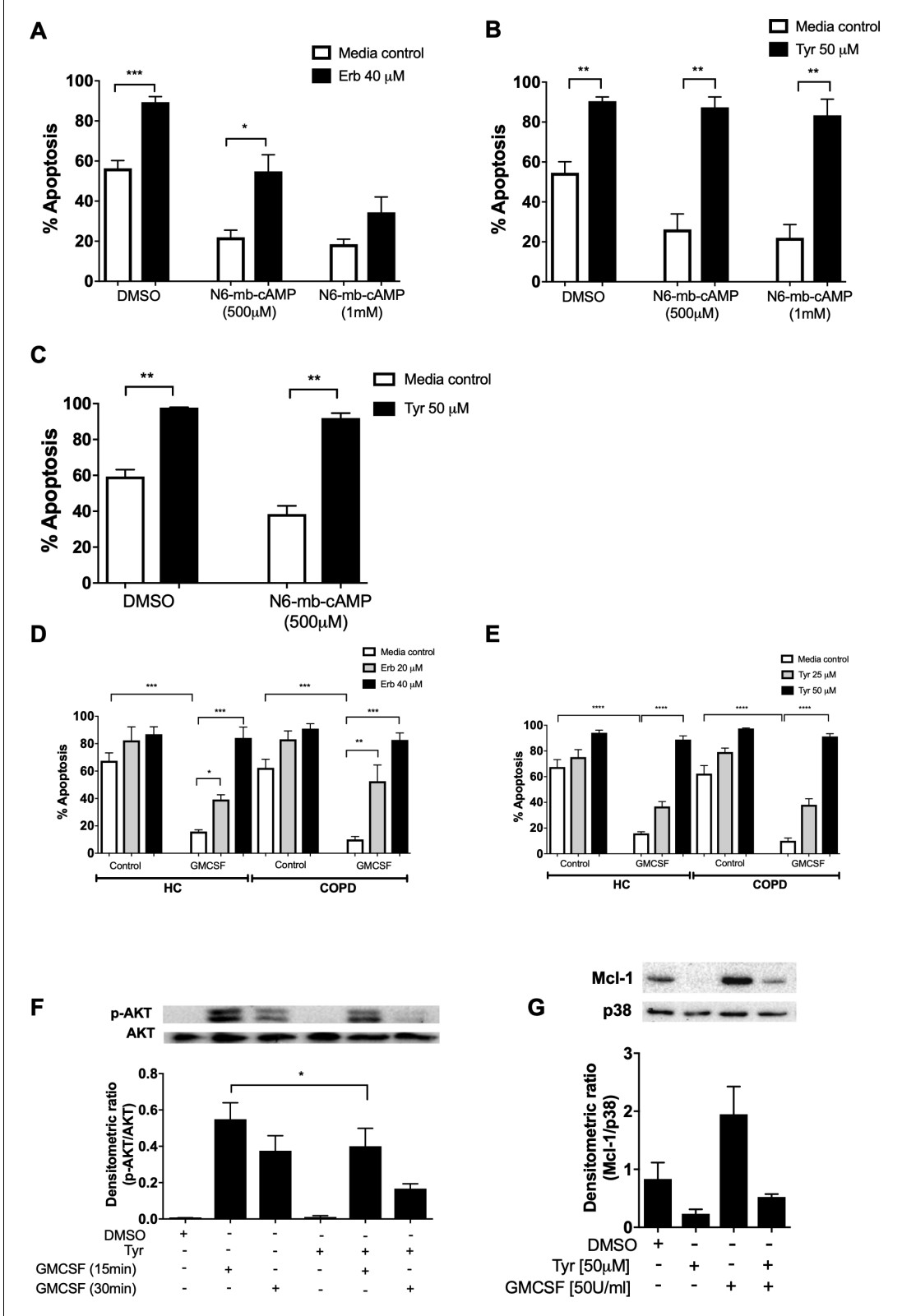

**Figure 3.** Erbstatin and tyrphostin AG825 overcome pro-survival effects of $N^6$-MB-cAMP and GMCSF. Neutrophils were incubated with DMSO, Erbstatin [Erb, 40 µM] (**A**) or tyrphostin AG825 [Tyr, 50 µM] (**B**) in the presence of DMSO or $N^6$-MB-cAMP [500 µM and 1 mM] for 20 hr. Neutrophils isolated from COPD patients were incubated with DMSO or tyrphostin AG825 [50 µM] in the presence of DMSO or $N^6$-MB-cAMP [500 µM] for 20 hr (**C**). Neutrophils isolated from COPD patients and age-matched healthy control subjects (HC) were incubated with DMSO, erbstatin (**D**) [20, 40 µM] or

*Figure 3 continued on next page*

*Figure 3 continued*

tyrphostin AG825 (**E**) [25, 50 µM] in the presence or absence of GMCSF [50 u/mL] for 20 hr. Apoptosis was assessed by light microscopy. The data are expressed as mean percentage apoptosis ± SEM from 4 to 6 independent experiments. (**F**) Neutrophils were incubated with DMSO or tyrphostin AG825 [Tyr, 50 µM] for 60 min before the addition of GMCSF [50 u/mL] for 15 or 30 mins. (**G**) Neutrophils were incubated with DMSO, tyrphostin AG825 [50 µM] for 60 min before the addition of GMCSF [50 u/mL] for a further 7 hr. Cells were lysed, subjected to SDS-PAGE electrophoresis and membranes probed for p-AKT, Mcl-1 or loading controls, AKT and P38. Images are representative of 3 independent experiments. Charts show densitometric values of 3 individual immunoblots and are expressed as a ratio of target (p-AKT or Mcl-1) over loading control (AKT or P38, respectively). Statistically significant differences were calculated by one-way ANOVA with Sidak post-test (**A–C, F–G**) or two-way ANOVA with Sidak post-test (**D–E**) and indicated as *$p < 0.05$, **$p < 0.01$, ***$p < 0.001$.

DOI: https://doi.org/10.7554/eLife.50990.008

## ErbB inhibitors and genetic knockdown increase apoptosis and reduce neutrophil number at the site of inflammation in vivo

To determine the ability of ErbB inhibition to exert an effect on neutrophil number and apoptosis in vivo, we used three complementary animal models of acute inflammation. To specifically address whether tyrphostin AG825 was able to accelerate apoptosis of neutrophils in the mammalian lung, we used a murine model of LPS-induced airway inflammation (*Thompson et al., 2014*). C57BL/6 mice nebulised with LPS developed an acute pulmonary neutrophilia after 48 hr, to a degree seen previously (*Figure 5A–B*) (*Thompson et al., 2014*). Tyrphostin AG825 had no effect on percentage of, or absolute number of neutrophils or macrophages compared to DMSO control (*Figure 5A–B*). Tyrphostin AG825 significantly increased the percentage of neutrophil apoptosis, both visualised as 'free' apoptotic cells (closed circles) and as a summation of both free apoptotic cells and apoptotic inclusions within macrophages in order to capture those that had been efferocytosed (closed triangles, *Figure 5C*). Macrophage efferocytosis was also significantly elevated by tyrphostin AG825,

**Table 1.** Kinexus antibody microarray analysis.

Ultrapurified neutrophils were incubated with $N^6$-MB-cAMP [100 µM] for 30 and 60 min or lysed immediately following isolation (0'). Lysates from four donors were pooled prior to Kinex antibody microarray analysis. Table shows all targets for which phospho-antibodies had Z ratios of >1.5 compared to t = 0 baseline control, at each timepoint. ErbB related antibodies are in bold.

| Target protein | Z-ratio (30' v 0') | Target protein | Z-ratio (60' v 0') |
|---|---|---|---|
| PDK1 | 5.69 | PDK1 | 4.79 |
| ZAP70/Syk | 4.85 | PKCa/b2 | 2.73 |
| p38a | 3.21 | Zap70/Syk | 2.72 |
| PLCg1 | 3.16 | p38a | 2.32 |
| MAP2K1 | 2.70 | S6Ka | 2.05 |
| FKHRL1 | 2.58 | Rb | 1.96 |
| GSK3a/b | 2.54 | PKCg | 1.79 |
| Huntingtin | 2.29 | **ErbB2** | **1.53** |
| BLNK | 2.25 | | |
| Jun | 1.99 | | |
| Rb | 1.99 | | |
| **ErbB2** | **1.94** | | |
| Btk | 1.92 | | |
| Bad | 1.81 | | |
| AMPKa1/2 | 1.70 | | |
| Synapsin 1 | 1.69 | | |
| PKBa | 1.64 | | |

DOI: https://doi.org/10.7554/eLife.50990.010

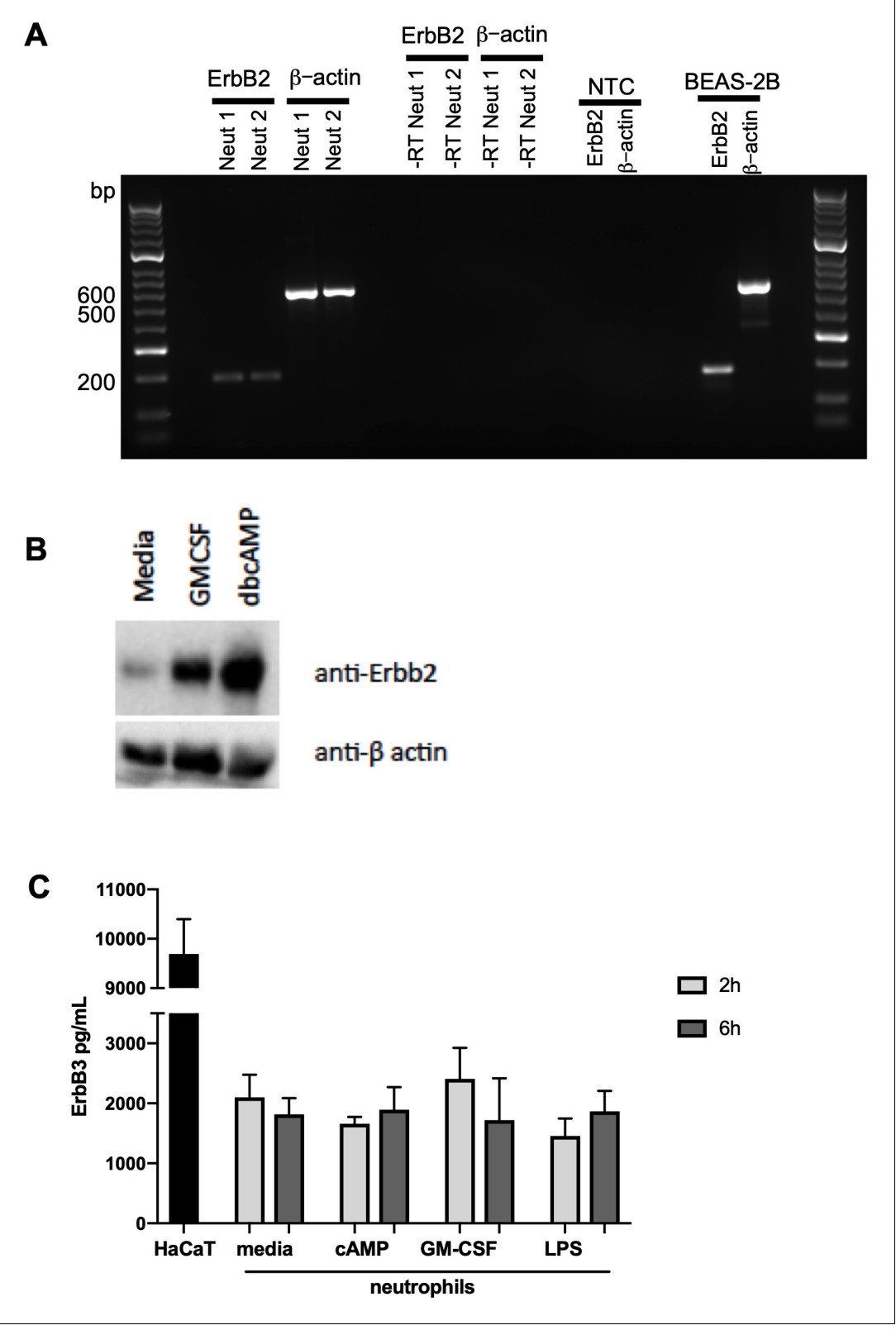

**Figure 4.** ErbB2 and ErbB3 expression and regulation in human neutrophils. (**A**) ErbB2 was detected in neutrophils and the positive control cell line, BEAS-2B, by RT-PCR. Primer sequences are as follows: ErbB2 forward: ACCCAGCTCTTTGAGGACAA, reverse: ATCGTGTCCTGGTAGCAGAG and β-actin forward: ATATCGCCGCGC TCGTCGTC, reverse: TAGCCGCGCTCGGTGAGGAT. NTC – no template control. (**B**) Neutrophils were treated

*Figure 4 continued on next page*

*Figure 4 continued*

with GMCSF [50 u/mL] and dbcAMP [10 μM] for 5 hr and lysates subjected to SDS PAGE. Membranes were immu-noblotted with antibodies to ErbB2 antibody or β-actin as a loading control. A 60kD band was detected which was upregulated by GMCSF and dbcAMP. The image is representative of three independent experiments. (C) ErbB3 was detected by ELISA in human neutrophils and the positive control cell line, HaCaT. Neutrophils were treated with media, dbcAMP [500 μM], GM-CSF [50 u/mL] or LPS [1 μg/mL] for 2 hr or 6 hr, after which lysates were col-lected and ELISA detecting total human ErbB3 was carried out (N = 4). Bars indicate mean + SEM and statistical differences between media control and treatments were measured by one-way ANOVA and Sidak post-test (C, ns).

DOI: https://doi.org/10.7554/eLife.50990.009

compared to vehicle control (*Figure 5D*), determined by counting the number of macrophages containing apoptotic inclusions as a proportion of total macrophages (*Figure 5E*). We next tested the anti-inflammatory potential of tyrphostin AG825 when administered once inflammation was established, which is more representative of the clinical scenario. Mice were i.p injected with zymosan to induce peritonitis and after 4 hr were treated (i.p.) with tyrphostin AG825 or vehicle control. Total cell counts in peritoneal lavage were $2.2 \times 10^6$ in PBS vs $1.7 \times 10^7$ in zymosan treated animals at 4 hr demonstrating established inflammation at this time point (*Navarro-Xavier et al., 2010*). Importantly, tyrphostin AG825 does not induce leukopenia (*Figure 5F*), however significantly fewer inflammatory cells were found in peritoneal lavage following tyrphostin AG825 treatment (*Figure 5G*). The neutrophil chemoattractant and proinflammatory cytokine, KC, was reduced in tyrphostin AG825 treated mice, and concomitant with this, a trend for less IL-6 was also observed (*Figure 5H*). IgM, which correlates with the number and activation of peritoneal B lymphocytes (*Almeida et al., 2001*), is significantly reduced in tyrphostin AG825-treated mice (*Figure 5I*).

To further extend this observation, we tested the ability of ErbB inhibitors to modulate neutro-philic inflammation resolution as a whole, in a model which encompasses multiple mechanisms of neutrophil removal including both apoptosis and reverse migration. In the *mpx*:GFP zebrafish tail fin injury model (*Renshaw et al., 2006*) (*Figure 6A*) we were able to show that tyrphostin AG825 (*Figure 6B*) and CP-724714 (*Figure 6C*) significantly reduced the number of neutrophils at the site of injury at 4 and 8 hpi. Simultaneous gene knockdown of *egfra* and *erbb2* via CRISPR/Cas9 (referred to as 'crispants') also recapitulated this phenotype (*Figure 6D*). Tyrphostin AG825 did not affect total neutrophil number (*Figure 6E*), but *egfra* and *erbb2* crispants had significantly fewer neutro-phils (*Figure 6F*). As demonstrated by TSA and TUNEL double staining (*Figure 6G*), tyrphostin AG825 upregulated neutrophil apoptosis at both the site of injury (*Figure 6H*) and in the caudal hematopoietic tissue (CHT) of zebrafish (*Figure 6I*). CHT neutrophil counts were unchanged between conditions (data not shown). *egfra* and *erbb2* crispants had increased numbers of apoptotic neutrophils at the site of injury, but this was not significant (*Figure 6J*), perhaps suggesting the presence of compensatory mechanisms. These findings show that inhibiting ErbB RTKs accelerate neutro-phil apoptosis in vitro and in vivo and enhance inflammation resolution, making ErbB inhibitors an attractive therapeutic strategy for inflammatory disease.

## Discussion

Neutrophils are powerful immune cells because of their destructive anti-microbial contents. A dele-terious by-product of this is their remarkable histotoxic potential to host tissue, ordinarily held in check by the onset of apoptosis. The inappropriate suppression of neutrophil apoptosis underpins a number of chronic inflammatory diseases, and we are yet to have available an effective treatment strategy that can reverse this cellular defect in clinical practice. Here we show in human, mouse and zebrafish models of inflammation and neutrophil cell death that targeting the ErbB family of RTKs regulates neutrophil survival and resolves inflammation.

Promoting neutrophil apoptosis is a desirable approach for the resolution of inflammation, since apoptosis functionally downregulates the cell, promotes rapid cell clearance by efferocytosis and engages an anti-inflammatory phenotype in phagocytosing cells (*Savill et al., 1989*; *Whyte et al., 1999*). As proof of principle, driving apoptosis experimentally promotes the resolution of inflamma-tion across multiple disease models (*Chello et al., 2007*; *Ren et al., 2008*; *Rossi et al., 2006*).

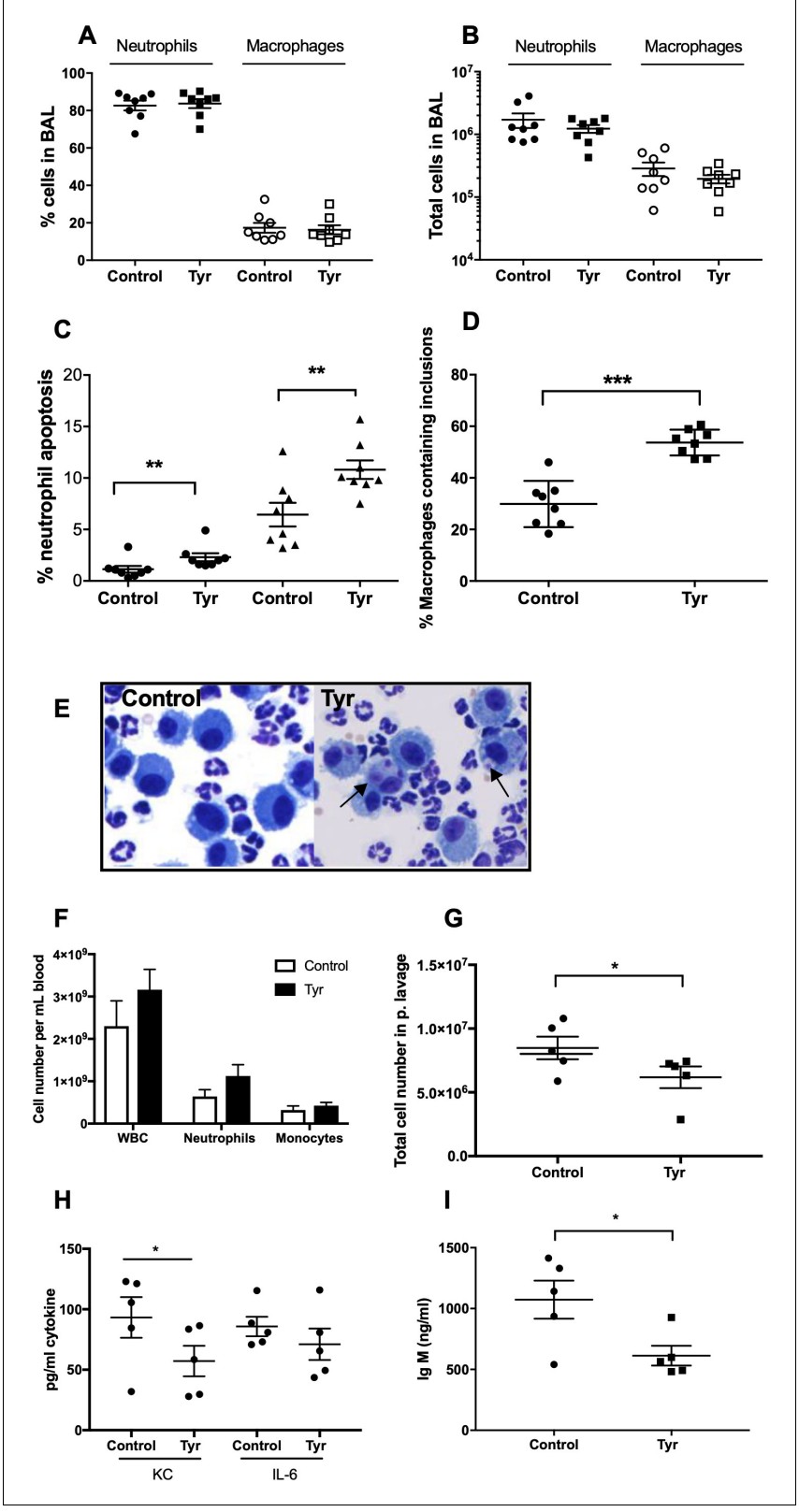

**Figure 5.** Tyrphostin AG825 increases neutrophil apoptosis and reduces inflammation in murine models of inflammation. C57BL/6 mice were nebulized with LPS and immediately injected intraperitoneally with either 10% DMSO (control, n = 8) or 20 mg/Kg tyrphostin AG825 (Tyr, n = 8). After 48 hr the mice were sacrificed and subjected to bronchoalveolar lavage. Percentage neutrophils (A, closed icons) and macrophages (A, open icons)

*Figure 5 continued on next page*

*Figure 5 continued*

and absolute numbers of neutrophils (B, closed icons) and macrophages (B, open icons) in BAL were calculated by haemocytometer and light microscopy. (**C**) Percentage neutrophil apoptosis (circles) and percentage neutrophil apoptosis calculated by also including numbers of apoptotic inclusions visualised within macrophages (triangles) was assessed by light microscopy. (**D**) Macrophages containing one or more apoptotic inclusions expressed as a percentage of all macrophages. Light microscopy image showing apoptotic inclusions within macrophages as indicated by black arrows (**E**). C57BL/6 mice were injected i.p. with 1 mg zymosan and 4 hr later injected i.p. with 20 mg/Kg tyrphostin AG825 (Tyr, n = 5) or 10% DMSO (Control, n = 5). At 20 hr mice were sacrificed and subjected to peritoneal lavage. (**F**) WBC, neutrophils and macrophages in blood were measured by a Sysmex cell counter. Total cells in peritoneal lavage were counted by flow cytometry (**G**) and KC, IL-6 (**H**) and IgM (**I**) measured in lavage by ELISA. At least two independent experimental replicates each processing 1–3 mice/group were performed. Statistical significance was calculated by Mann–Whitney U test (**A–D and G–I**) or one-way ANOVA with Sidak post-test (**F**) and indicated as $*p<0.05$, $**p<0.01$, $***p<0.001$.

DOI: https://doi.org/10.7554/eLife.50990.011

Several compounds targeting the ErbB family have been approved as medicines for the treatment of cancer (*Singh et al., 2016*). Our findings open up the possibility of repurposing well-tolerated ErbB inhibitors for patients with inflammatory disease, potentially addressing a currently unmet clinical need.

The ErbB family are critical regulators of cell proliferation and are associated with the development of many human malignancies (*Roskoski, 2014*). In addition to the development of cancer, ErbB members have known roles in inflammatory diseases of the airway, skin and gut (*Davies et al., 1999*; *Finigan et al., 2011*; *Frey and Brent Polk, 2014*; *Hamilton et al., 2003*; *Pastore et al., 2008*). In the context of lung inflammation, ErbB2 is upregulated in whole lung lysates in murine bleomycin models of lung injury and EGFR ligands are increased in BAL from acute lung injury patients receiving mechanical ventilation (*Finigan et al., 2011*), suggesting ErbB signalling axes may play a role in the process of airway inflammation in vivo. We show, in murine models where Tyrphostin AG825 was administered either at the time of inflammatory stimulus or once inflammation was established, an impact on cell number, proinflammatory cytokine production and neutrophil apoptosis, further validating the use of ErbB inhibitors to reduce inflammation. The benefit of EGFR inhibitors in reducing inflammation in ventilator-induced and OVA/LPS-induced lung injury rodent models is shown by others, further supporting the targeting of this pathway in inflammatory disease settings (*Bierman et al., 2008*; *Shimizu et al., 2018*; *Takezawa et al., 2016*).

Others have reported that neutrophils express members of the ErbB family (*Lewkowicz et al., 2005*), particularly ErbB2 at low levels (*Petryszak et al., 2016*) and we show that they are phosphorylated and regulated following exposure to inflammatory stimuli. ErbBs have known roles in suppressing apoptosis of epithelial cells and keratinocytes, but this study is the first to show a role for ErbBs in survival signalling of myeloid cells. Little is known about the roles of ErbBs in neutrophil function. Erbstatin has been shown to inhibit neutrophil ROS production (*Dreiem et al., 2003*; *Mócsai et al., 1997*; *Reistad et al., 2005*) and chemotactic responses (*Yasui et al., 1994*). Other kinase families have been found to play a role in neutrophil survival and neutrophilic inflammation, most notably the cyclin-dependent kinases (CDKs) (*Rossi et al., 2006*). In accordance with this, compounds targeting CDKs were identified as drivers of neutrophil apoptosis in both our primary and secondary screens. Moreover, p38 MAPK inhibitor compounds were also identified in both zebrafish and human screens, and since this kinase is known to mediate survival signals, these findings give confidence to the robustness of the screen design and execution.

The engagement of apoptosis by the ErbB inhibitors erbstatin and tyrphostin AG825 was confirmed both biochemically by phosphatidylserine exposure, and mechanistically by the caspase inhibitor Q-VD-OPh and loss of Mcl-1. This suggests that inhibiting ErbBs as a therapeutic strategy may achieve an overall anti-inflammatory effect in vivo systems, facilitating clearance by macrophages. In support of this, we provide evidence of increased efferocytosis in vivo following tyrphostin AG825 treatment, with no evidence of secondary neutrophil necrosis due to overwhelming macrophage clearance capacity, evidenced both morphologically and by TO-PRO-3 staining.

The ability of ErbB inhibitors to promote neutrophil apoptosis even in the presence of multiple pro-survival stimuli emphasises the potential of ErbB inhibitors in the lung, at sites where

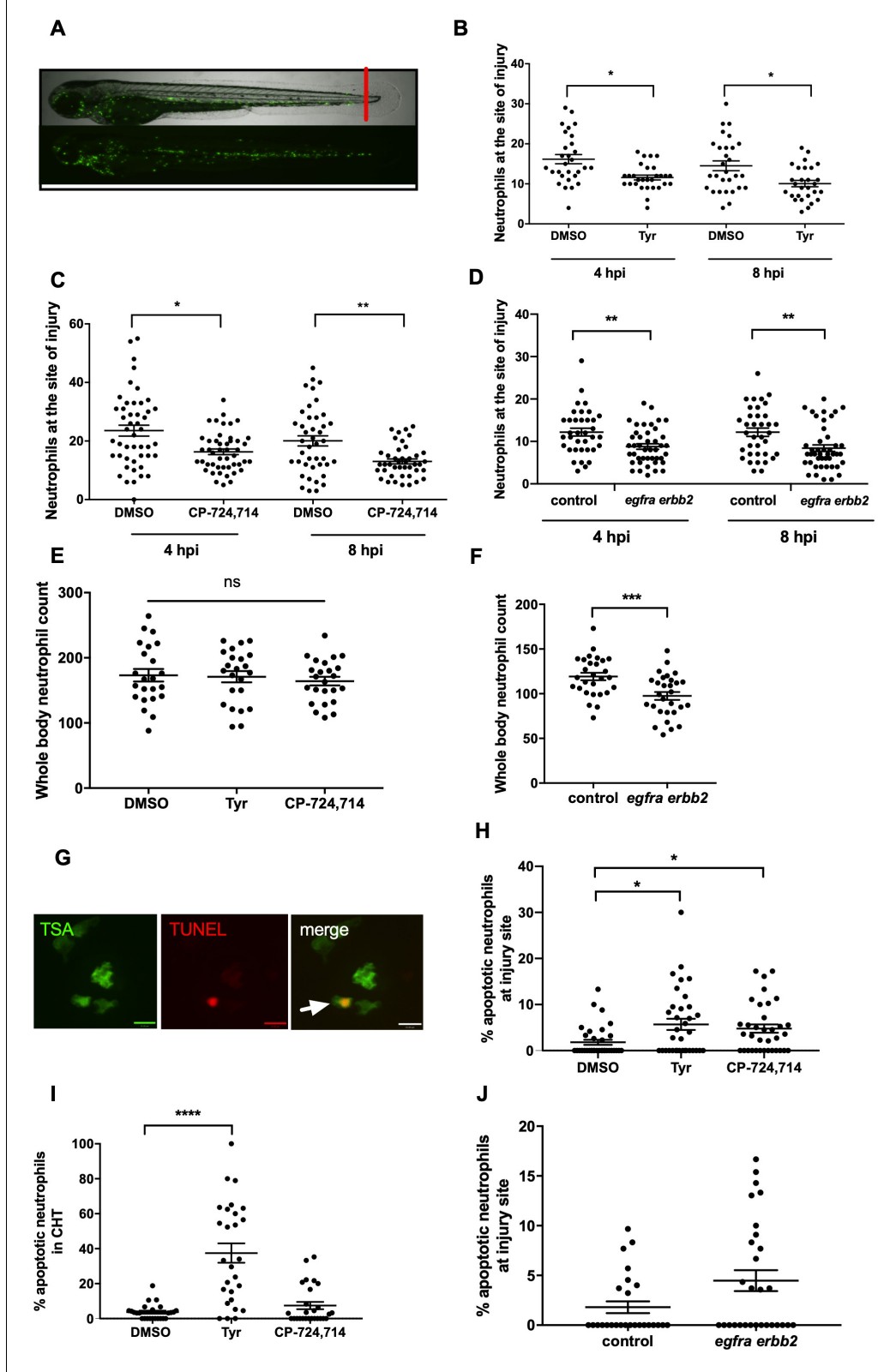

**Figure 6.** Pharmacological inhibition and genetic knockdown of *egfra* and *erbb2* by CRISPR/Cas9 reduces neutrophil number at the site of injury in a zebrafish model of inflammation. Tail fin transection was performed as indicated by the red line (**A**, upper image). Zebrafish larvae (*mpx*:GFP) were pre-treated at two dpf with DMSO, tyrphostin AG825 [Tyr, 10 μM] (**B**, minimum n = 28 larvae per condition), or CP-724714 [10 μM] (**C**, minimum n = 42 larvae per condition) for 16 hr followed by injury. *egfra* and *erbb2* crispants were generated and injured at two dpf (**D**, minimum n = 36 larvae per

*Figure 6 continued on next page*

*Figure 6 continued*

condition). The number of neutrophils at the site of injury was determined at 4 and 8 hpi by counting GFP-positive neutrophils. To enumerate neutrophils across the whole body, uninjured inhibitor treated larvae (three dpf) (**E**, minimum n = 23 larvae per condition) or crispants (two dpf) (**F**, minimum n = 28 larvae per condition) were imaged by fluorescent microscopy (**A**, lower image). Apoptosis was measured at the site of injury after 8 hr by TSA and TUNEL double staining (**G**) (white arrow indicates TUNEL positive neutrophil, scale bar 10 μM) of mpx:*GFP* tyrphostin AG825 [Tyr, 10 μM] or CP-724714 [10 μM] treated larvae at three dpf (**H**, minimum n = 35 larvae per condition). Uninjured inhibitor treated larvae were assessed for neutrophil apoptosis in the CHT at three dpf (**I**, minimum n = 27 larvae per group). Apoptosis at the tail fin injury site of *egfra erbb2* crispants at two dpf was also measured at eight hpi (**J**, minimum n = 26 larvae per group). All data collated from at least three independent experiments, displayed as mean ± SEM. Each icon shows one data point from one individual larvae. Statistically significant differences were calculated by two-way ANOVA with Sidak post-test (**B–D**) or one-way ANOVA with Dunnett's post-test(**E**), Students' t-test (**F**), Kruskal-Wallis test with Dunn's post-test (**H–I**) or Mann-Whitney U test (**J**), and indicated as *p<0.05, **p<0.01, ***p<0.001, ****p<0.0001.

DOI: https://doi.org/10.7554/eLife.50990.012

inflammatory mediators are in abundance and where neutrophils are exposed to microorganisms. This is supported by the ability of tyrphostin AG825 to prevent early pro-survival signalling in response to GMCSF, including the phosphorylation of AKT. This precedes the onset of apoptosis, occurring at a time point (15 min) where apoptosis is typically less than 1%. Others have shown the ability of erbstatin to prevent GMCSF-mediated activation of PI3K in human neutrophils, although the impact on cell survival was not studied (*al-Shami et al., 1997*). Therefore, ErbBs may function as an early and upstream component of the survival pathway in neutrophils. Subsequent impact on Mcl-1 destabilisation by tyrphostin AG825 at 8 hr suggests a cellular mechanism by which these pro-apoptotic effects are mediated.

The effects of ErbB inhibitors in driving spontaneous apoptosis suggest that, under certain circumstances, ErbB activity might be required for constitutive neutrophil survival. It is not clear what, if anything, engages ErbB signalling in culture. The rapid phosphorylation of ErbB2 following $N^6$-MB-cAMP treatment (30 min) suggests that perhaps a ligand is not required, or that the neutrophils can rapidly release ErbB agonists in an autocrine manner. Unlike all other ErbBs, ErbB2 monomers exist in a constitutively active conformation and can form homodimers that do not require a ligand for activation (*Fan et al., 2008*). ErbBs achieve great signalling diversity: in part because of the individual biochemical properties of ligands and multiple homo-heterodimer combinations, and in part because they activate multiple components including those known to be critical in neutrophil cell survival such as PI3K, MAPK and GSK-3, as well as phosphorylating the Bcl-2 protein Bad which inhibits its death-promoting activity (*Yarden and Sliwkowski, 2001*).

A limitation of our study is the genetically intractability of human neutrophils, meaning we cannot exclude the possibility that the inhibitors are having off target effects in this system. Mammalian models of ErbB deletion are limited by profound abnormalities in utero and during development (*Britsch et al., 1998*; *Dackor et al., 2007*; *Gassmann et al., 1995*; *Miettinen et al., 1995*; *Riethmacher et al., 1997*). For this reason, CRISPR/Cas9 was used to knockdown *egfra* and *erbb2* in zebrafish, which confirmed a role for ErbBs in resolving inflammation. Targeting ErbBs genetically and pharmacologically reduces the number of neutrophils at the site of injury in zebrafish, which may reflect inhibition of a number of pathways that regulate neutrophil number in the tissue, including migration pathways (*Ellett et al., 2015*). However, the increase in apoptotic neutrophil count at the site of injury with ErbB inhibitor treatment suggests ErbBs may be inducing anti-apoptotic signalling pathways within this inflammatory environment, which could at least in part be causing the phenotype. The reduced neutrophil count at the injury site may also be due to the increase in apoptotic neutrophils in the CHT, which may be preventing neutrophil migration to sites of injury. The unchanged whole body neutrophil number is potentially due to compensatory upregulation of neutrophil production within the CHT. Genetic deletion, but not pharmacological inhibition, of *egfra* and *erbb2* significantly reduced whole body neutrophil number, which may reflect crispants being without *egfra* and *erbb2* genes from a one-cell stage. Reduced neutrophils at the injury site of crispants may be explained by their reduced whole body neutrophil number, but potentially also defects in the migratory response of these neutrophils to a site of inflammation. Murine models of inflammatory disease, where tyrphostin AG825 was administered either at the time of inflammatory stimulus or once inflammation was established, show an impact on cell number, proinflammatory cytokine

production and neutrophil apoptosis, further validating the use of ErbB inhibitors to reduce inflammation.

In conclusion, we have identified a previously undefined role for ErbB RTKs in neutrophil survival pathways and a potential new use for ErbB inhibitors in accelerating inflammation resolution. These findings suggest the ErbB family of kinases may be novel targets for treatments of chronic inflammatory disease, and the potential for repurposing ErbB inhibitors currently in use for cancer may have significant clinical potential in a broader range of indications.

## Materials and methods

### Experimental design

Our objectives for this study are to identify compounds that are able to resolve neutrophilic inflammation. To do this we performed unbiased chemical screens in both human neutrophils in vitro and zebrafish models of inflammation in vivo. Results were validated in murine models of peritoneal and airway inflammation and zebrafish tail injury models. Genetic evidence was obtained by CRISPR/Cas9 genetic editing in zebrafish.

### Isolation and culture of human neutrophils

Neutrophils were isolated from peripheral blood of healthy subjects and COPD patients by dextran sedimentation and discontinuous plasma-Percoll gradient centrifugation, as previously described (*Haslett et al., 1985*; *Ward et al., 1999b*) in compliance with the guidelines of the South Sheffield Research Ethics Committee (for young healthy subjects; reference number: STH13927) and the National Research Ethics Service (NRES) Committee Yorkshire and the Humber (for COPD and age-matched healthy subjects; reference number: 10/H1016/25). Informed consent was obtained after the nature and possible consequences of the study were explained. Mean age in years was $61.7 \pm 2.3$ (n = 10) and $66.0 \pm 3.6$ (n = 7) for COPD and age-matched healthy subjects respectively. Ultrapure neutrophils, for Kinexus antibody array experiments, were obtained by immunomagnetic negative selection as previously described (*Sabroe et al., 2002*). Neutrophils were cultured ($2.5 \times 10^6$/ml) in RPMI 1640 (Gibco, Invitrogen Ltd) supplemented with 10% FCS 1% penicillin-streptomycin, in the presence or absence of the following reagents: GMCSF (PeproTech, Inc), $N^6$-MB-cAMP (Biolog), anti-ErbB3 blocking antibody, Tyrphostin AG825 (both Sigma-Aldrich), CP-724714 (AdooQ Bioscience), Erbstatin analog (Cayman Chemicals), Pyocyanin (*Usher et al., 2002*) or compounds from PKIS (Published Kinase Inhibitor Set 1, GlaxoSmithKline) at concentrations as indicated.

In vitro screening of PKIS in neutrophil apoptosis assays. PKIS consists of 367 small molecule protein kinase inhibitors and is profiled with respect to target specificity (*Elkins et al., 2016*). In primary screen experiments, neutrophils (from five independent donors over 5 days) were incubated with each compound at 62 µM for 6 hr. Apoptosis was measured by flow cytometry (Attune, Invitrogen). Secondary screening was performed with selected compounds that accelerated neutrophil apoptosis greater than twofold in the primary screen. Compounds were incubated with neutrophils at 10 µM for 6 hr and apoptosis assessed by Attune flow cytometry.

### Human neutrophil apoptosis assays

Neutrophil apoptosis was assessed by light microscopy and by flow cytometry. Briefly, for the assessment of apoptosis by light microscopy based on well-characterised morphological changes, neutrophils were cytocentrifuged, fixed with methanol, stained with Reastain Quick-Diff (Gentaur), and then apoptotic and non-apoptotic neutrophils were counted with an inverted, oil immersion microscope (Nikon Eclipse TE300, Japan) at 100X magnification (*Savill et al., 1989*). To assess apoptosis by flow cytometry, neutrophils were stained with PE conjugated Annexin-V (BD Pharminogen) and TO-PRO-3 (Thermofisher Scientific) (*Savill et al., 1989*; *Vermes et al., 1995*; *Ward et al., 1999a*) and sample acquisition was performed by an Attune flow cytometer (Life Technologies) and data analysed by FlowJo (FlowJo LLC).

### Kinexus antibody array

Neutrophils were incubated with $N^6$-MB-cAMP [100 µM] for 30 and 60 min or lysed immediately following isolation (t0). Cells were lysed in PBS containing Triton-X, 1 µM PMSF and protease inhibitor

cocktail and following 2 min on ice were centrifuged at 10,000 RPM to remove insoluble material. Lysates (containing protein at 6 mg/mL) from four donors were pooled prior to Kinex antibody microarray analysis (Kinexus Bioinformatics) (*Zhang and Pelech, 2012*). Lysates are subjected to 812 antibodies including phospho-site specific antibodies to specifically measure phosphorylation of the target protein. Fluorescent signals from the array were corrected to background and log2 transformed and a Z score calculated by subtracting the overall average intensity of all spots within a sample, from the raw intensity for each spot, and dividing it by the standard deviations (SD) of all the measured intensities within each sample (*Cheadle et al., 2003*). Z ratio values are further calculated by taking the difference between the averages of the Z scores and dividing by the SD of all differences of the comparison (e.g, 30 min treated samples versus 0 min control). A Z ratio of ±1.5 is considered to be a significant change from control.

## Western blotting

Whole cell lysates were prepared by resuspending human neutrophils ($5 \times 10^6$) in 50 µl hypotonic lysis buffer (1 mM PMSF, 50 mM NaF, 10 mM Sodium orthovanadate, protease inhibitors cocktail in water), and by boiling with 50 µl 2X SDS buffer (0.1M 1,4-Dithio-DL-threitol, 4% SDS, 20% Glycerol, 0.0625M Tris-HCl pH6.8% and 0.004% Bromophenol blue). Protein samples were separated by SDS-polyacrylamide gel electrophoresis, and electrotransfer onto PVDF (polyvenylidene difluoride) membranes was performed by semi-dry blotting method. Membranes were then blocked with 5% skimmed milk in TBS-tween and probed against antibodies to p-AKT, AKT (both Cell signalling Technology), Mcl-1 (Santa Cruz Biotechnology), ErbB2 (New England Biolabs), p38 or β-actin (loading controls, StressMarq Biosciences Inc or Sigma respectively), followed by HRP-conjugated secondary antibodies and detection with chemiluminescent substrate solution ECL2 (GE Healthcare).

## Fish husbandry

The neutrophil-specific, GFP-expressing transgenic zebrafish line, *Tg(mpx:GFP)i114*, (referred to as *mpx*:GFP) (*Renshaw et al., 2006*) was raised and maintained according to standard protocols (*Nüsslein-Volhard and Dahm, 2002*) in UK Home Office approved aquaria in the Bateson Centre at the University of Sheffield, according to institutional guidelines. Adult fish are maintained in 14 hr light and 10 hr dark cycle at 28°C.

## Zebrafish tail injury model of inflammation

*PKIS screening*: Tail fin transection was performed on *mpx*:GFP zebrafish larvae at 3 days post-fertilisation (dpf) (*Elks et al., 2011*; *Renshaw et al., 2006*). At 6 hr post-injury (hpi), larvae that had mounted a good inflammatory response, as defined by recruitment of >15 neutrophils to the injury site, were arrayed at a density of 3 larvae per well and incubated with PKIS compounds at a final concentration of 25 µM or vehicle control for a further 6 hr. At 12 hpi, the plate was scanned using prototype PhenoSight equipment (Ash Biotech). Images were scored manually as described previously (*Robertson et al., 2014*). In brief, each well of three larvae was assigned a score between 0–3, corresponding to the number of larvae within the well with a reduced number of neutrophils at the site of injury. Kinase inhibitors which reduced green fluorescence at the injury site to an extent that their mean score was ≥1.5 were regarded as hit compounds.

*ErbB inhibition studies:* Briefly, two dpf *mpx*:GFP larvae were treated with Tyrphostin AG825 [10 µM] for 16 hr before undergoing tailfin transection (*Elks et al., 2011*; *Renshaw et al., 2006*). The number of neutrophils at the site of injury was determined at 4 and 8 hpi by counting GFP-positive neutrophils by fluorescent microscopy. To enumerate neutrophils across the whole body, uninjured larvae were treated with Tyrphostin AG825 [10 µM] for 24 hr and then mounted in 0.8% low-melting point agarose (Sigma-Aldrich) followed by imaging by fluorescence microscopy (Nikon Eclipse TE2000-U) at 4X magnification, followed by manual counting.

## Zebrafish apoptosis assays

Larvae from each experimental group were pooled into 1.5 mL eppendorf tubes. TSA signal amplification of GFP-labelled neutrophils (driven by endogenous peroxidase activity) was carried out using TSA Plus Fluorescein System (Perkin Elmer). Larvae were fixed overnight in 4% paraformaldehyde at 4°C after which they were subjected to proteinase K digestion. Larvae were post-fixed in 4%

paraformaldehyde, before subsequent TUNEL staining for apoptosis using ApopTag Red In Situ Apoptosis Detection Kit (Millipore). Larvae were then mounted in low-melting point agarose and images acquired and analysed using UltraVIEWVoX spinning disc confocal laser imaging system with Volocity 6.3 software (Perkin Elmer). Apoptotic neutrophil count was determined firstly by identifying cells with co-localisation of the TSA and TUNEL stains, then confirmed by accounting for apoptotic neutrophil morphology.

## Generation of transient CRISPR/Cas9 zebrafish mutants

Transient dual knockdown of *egfra* and *erbb2* was induced using a Cas9 nuclease (New England Biolabs) in combination with transactivating RNA (tracr) and synthetic guide RNAs specific to zebrafish *egfra* and *erbb2* genes (Merck). The non-targetting control in these experiments was a guide RNA targetted towards *tyrosinase*, a gene involved in pigment formation and therefore easy to identify when mutated, and which is used by others in the field as a CRISPR/Cas9 control (*O'Connor et al., 2019*; *Varshney et al., 2016*). We have previously shown that this guide does not influence neutrophilic inflammation in the zebrafish (*Evans et al., 2019*; *Isles et al., 2019*). Guide RNAs were designed using the online tool CHOPCHOP (https://chopchop.cbu.uib.no/) with the following sequences: *efgra*: TGAATCTCGGAGCGCGCAGGAGG; *erbb2*: AACGCTTTGGACCTACACG TGGG; *tyrosinase*: GGACUGGAGGACUUCUGGGG. Each guide RNA was resuspended to 20 µM in nuclease-free water with 10 mM Tris-HCl (pH8). Guide RNA [20 µM], tracr [20 µM] and Cas9 protein [20 µM] were combined (in a 1:1:1 ratio). 0.5 µL phenol red was added to each injection solution for visualisation. A graticule was used to calibrate glass capillary needles to dispense 0.5 nL of injection solution, and 1 nL was injected into the yolk sac of single-cell stage *mpx*:GFP embryos. Tail injury assays were carried out at two dpf as described above.

## Genotyping of crispant larvae

High-resolution melt curve analysis was used to determine the rate of *egfra* and *erbb2* mutation in larvae at two dpf. Genomic DNA was collected from individual larvae in both the control and experimental groups, by adding 90 µL 50 mM NaOH to each larvae in a 96-well qPCR plate and incubating at 95°C for 20 min. 10 µL Tris-HCl (pH 8) was then added as a buffer. Master mixes containing either *egfra* or *erbb2* primers (Integrated DNA Technologies) (sequences in table below) were made up, with each well containing: 0.5 µL 10 µM forward primer, 0.5 µL 10 µM reverse primer, 5 µL 2X DyNAmo Flash SYBR Green (Thermo Scientific), 3 µL milliQ water. One µL genomic DNA was added to each master mix in a 96-well qPCR plate. Melt curve analysis was performed and analysed with Bio-Rad Precision Melt Analysis software. Mutation rate was calculated by determining the percentage of *egfra erbb2* larvae that showed a different melt-curve profile to the genomic DNA collected from *tyrosinase* fish (based on 95% confidence intervals). The average mutation rate in our experiments was 97.5% and 87.9% for *egfra* and *erbb2*, respectively.

Primer sequences used for high-resolution melt curve analysis.

| Gene | Forward primer sequence | Reverse primer sequence | Product size |
| --- | --- | --- | --- |
| *egfra* | CCAGCGGTTCGGTTTATTCAG | CGTCTTCGCGTATTCTTGAGG | 100 |
| *erbb2* | ACAAAGAGCCCAAAAACAGGTTTA | TCCTTCAGTGCATACCCAGA | 93 |

## Murine model of LPS induced acute lung inflammation

All work involving animals was performed in accordance with the Animal (Scientific procedures) Act 1986. Protocols were produced in line with PREPARE guidelines and FRAME recommendations and were reviewed by the University of Sheffield's Animal Welfare Committee. C57BL/6 mice (female, 9–10 weeks old) were nebulised with LPS (3 mg per group of 8 mice) (*Pseudomonas aeruginosa*, Sigma-Aldrich) and immediately injected intraperitoneally (i.p.) with either Tyrphostin AG825 (Tocris Bioscience) at 20 mg/Kg in 10% DMSO v/v in vegetable oil (eight mice, treatment group) or an equivalent volume of 10% DMSO v/v in vegetable oil (eight mice, control group) (*Kedrin et al., 2009*; *Roos et al., 2014*). After 48 hr the mice were sacrificed by terminal anaesthesia by i.p. pentobarbitone and subjected to bronchoalveolar lavage (BAL, 4 × 1 mL of saline). BAL samples were microcentrifuged and the cellular fraction counted by a hemocytometer and cytocentrifuged.

Neutrophil apoptosis and macrophage efferocytosis of apoptotic neutrophils was quantified by oil immersion light microscopy (Nikon Eclipse TE300, Japan).

## Murine model of zymosan-induced peritonitis

C57BL/6 mice were i.p. injected with 1 mg zymosan (Sigma-Aldrich) and 4 hr later injected i.p with 20 mg/Kg Tyrphostin AG825 in 10% DMSO v/v in vegetable oil (five mice, treatment group) or an equivalent volume of 10% DMSO v/v in vegetable oil (five mice, control group). At 20 hr the mice were subjected to terminal gaseous anaesthesia (isoflurane) followed by a cardiac puncture and peritoneal lavage (4 × 1 mL of saline). WBC, neutrophils and macrophages were enumerated in blood by an automated haematology analyser (KX-21N, Sysmex, Milton Keynes, UK). Lavage samples were microcentrifuged and the cellular fraction subjected to flow cytometry and cytocentrifuged for light microscopy. IL-6, KC (Duoset ELISA kits, R and D systems) and IgM (Thermofisher Scientific) in cell free lavage were measured by ELISA as per manufacturer's instructions.

## Statistical analysis

Data were analysed using GraphPad Prism 8 (GraphPad Software, San Diego, CA) using one-way or two-way ANOVA (with appropriate post-test detailed in the Figure legends) for all in vitro data and appropriate in vivo experiments. Non-parametric tests (Mann-Whitney U-test or Kruskal-Wallis test) were used for selected in vivo experiments with non-Gaussian distribution. Data are expressed as mean ± SEM (standard error of mean), and significance was accepted at $p < 0.05$.

## Acknowledgements

**General**: We thank Lynne Williams, Carl Wright, Jessica Willis, Elizabeth Marsh and Catherine Loynes for help with animal experiments as well as volunteers and patients who donated blood to this study. We thank the Bateson Centre aquaria staff for their assistance with zebrafish husbandry.

## Additional information

### Competing interests

Caroline O Savage: is an employee of GlaxoSmithKline Research and Development Ltd. The author declares no other competing interests exist. The other authors declare that no competing interests exist.

### Funding

| Funder | Grant reference number | Author |
| --- | --- | --- |
| Commonwealth Scholarship Commission | | Atiqur Rahman |
| Medical Research Council | MR/M004864/1 | Stephen A Renshaw |
| Medical Research Council | G0700091 | Stephen A Renshaw |
| European Commission | PITG-GA-2011-289209 | Julien JY Rougeot Annemarie H Meijer |
| SGC | | William J Zuercher |
| British Heart Foundation | Intermediate Clinician Fellowship FS/18/13/33281 | Abigail E Reese |

The funders had no role in study design, data collection and interpretation, or the decision to submit the work for publication.

### Author contributions

Atiqur Rahman, Investigation, Writing—original draft, Writing—review and editing; Katherine M Henry, Conceptualization, Formal analysis, Investigation, Writing—original draft, Writing—review and editing; Kimberly D Herman, Formal analysis, Investigation, Writing—review and editing; Alfred

AR Thompson, Hannah M Isles, Nika Khoshaein, Abigail E Reese, Kathryn Higgins, Caroline Tabor, Investigation, Writing—review and editing; Claudia Tulotta, Formal analysis, Investigation, Reviewing and editing draft; David Sammut, Resources, Investigation, Writing—review and editing; Julien JY Rougeot, Funding acquisition, Formal analysis; Ian Sabroe, Conceptualization, Formal analysis, Funding acquisition, Writing—review and editing; William J Zuercher, Conceptualization, Resources, Funding acquisition, Writing—review and editing; Caroline O Savage, Writing—review and editing, Was key to the development of ideas that led on to this work; Annemarie H Meijer, Conceptualisation, Funding acquisition, Writing—review and editing; Moira KB Whyte, Conceptualization, Formal analysis, Writing—review and editing; David H Dockrell, Conceptualization, Formal analysis, Supervision, Writing—review and editing; Stephen A Renshaw, Conceptualization, Resources, Formal analysis, Supervision, Funding acquisition, Writing—original draft, Project administration, Writing—review and editing; Lynne R Prince, Conceptualization, Resources, Formal analysis, Supervision, Writing—original draft, Project administration, Writing—review and editing

### Author ORCIDs
Katherine M Henry (iD) http://orcid.org/0000-0003-0554-2063
Alfred AR Thompson (iD) https://orcid.org/0000-0002-0717-4551
Stephen A Renshaw (iD) https://orcid.org/0000-0003-1790-1641
Lynne R Prince (iD) https://orcid.org/0000-0001-6133-9372

### Ethics

Human subjects: Peripheral blood of healthy subjects and COPD patients was taken following informed consent and in compliance with the guidelines of the South Sheffield Research Ethics Committee (for young healthy subjects; reference number: STH13927) and the National Research Ethics Service (NRES) Committee Yorkshire and the Humber (for COPD and age-matched healthy subjects; reference number: 10/H1016/25).

Animal experimentation: Zebrafish were raised and maintained according to standard protocols in UK Home Office approved aquaria in the Bateson Centre at the University of Sheffield, according to institutional guidelines. All work involving mice was performed in accordance with the Animal (Scientific procedures) Act 1986 and has been approved by the Animal welfare and ethical review body at University of Sheffield. Work was carried out under procedure project license 40/3726. All animals were checked prior to the start of experiments by competent personal licensees (PIL), and were deemed to be fit and well before the start of experiments.

### Decision letter and Author response
Decision letter https://doi.org/10.7554/eLife.50990.016
Author response https://doi.org/10.7554/eLife.50990.017

## Additional files

### Supplementary files
• Supplementary file 1. PKIS compounds that accelerated neutrophil apoptosis >2 fold over control. PKIS compounds were incubated with neutrophils for 6 hr and apoptosis was assessed by Annexin V/TO-PRO-3 staining by flow cytometry. Sixty-two compounds accelerated apoptosis $\geq$2 fold and compound names are presented here, along with fold change over control. The kinase profiling information for each of these inhibitors is available to download at https://www.nature.com/articles/nbt.3374#supplementary-information (file 'PKIS Nanosyn Assay Heatmaps' from Supplementary Data) (*Elkins et al., 2016*).
DOI: https://doi.org/10.7554/eLife.50990.013
• Transparent reporting form DOI: https://doi.org/10.7554/eLife.50990.014

### Data availability
All data generated or analysed during this study are included in the manuscript and supporting files.

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
