## [Decision Letter]

[Editors’ note: a previous version of this study was rejected after peer review, but the authors submitted for reconsideration. The first decision letter after peer review is shown below.]

Thank you for submitting your work entitled "Defining the functional neutrophil kinome reveals ErbB kinases as potential therapeutic targets in inflammatory disease" for consideration by *eLife*. Your article has been reviewed by a Senior Editor, a Reviewing Editor, and three reviewers. The following individuals involved in review of your submission have agreed to reveal their identity: Jean-Pierre Levraud.

Our decision has been reached after consultation between the reviewers. Based on these discussions and the individual reviews below, we regret to inform you that your work will not be considered further for publication in *eLife*.

Within our discussion, the reviewers all felt that while the story on the ErbB2 role in inflammation was interesting, more work was needed to really conclude its specific role, especially as this is not the first study to identify a role for ErbBs in neutrophil function and inflammation. Moreover, there was a consensus that the manuscript, as it stands now, really consists of two incomplete and disconnected stories: the kinome and the inhibitor screen. They were also more puzzled that ErB2, which was the main focus of the paper, did not even come up in the kinome. This brings up questions about both approaches. Overall, the impression was that this paper would be better understood as two separate stories but that both stories needed more experiments to be convincing, as outlined in the specific comments, below. One of the reviewers points out that there is a conditional (floxed) Erb2 mutant in mouse (Ozcelik et al., 2002). If the authors use this line and a neutrophil-specific Cre line, they could add genetic evidence. They also agreed that kinase assays needed to be performed to show that the effects were specific as well as live imaging of their zebrafish experiments to determine whether the effects were on cell migration or cell death. Because we all felt that it was unlikely for you produce more convincing data in less than 2 months, the usual turnaround for *eLife*, any resubmission would need to be as a new paper, and we would argue without the kinome data.

Reviewer #1:

The authors report that inhibition of ErbB kinase limits neutrophil inflammation by affecting neutrophil apoptosis. The paper uses several systems: human neutrophils, zebrafish and mice to probe the effect of ErbB kinase inhibition on inflammation. The observations are intriguing and suggest that ErbB inhibition may provide a therapeutic target for inflammatory disease.

Essential revisions:

Genetic approaches to implicate ErbB kinases in these processes, especially in the zebrafish model or cell lines would better support their conclusion that ErbB kinases regulate neutrophil apoptosis and inflammation. Are there genetic mutants or knockdowns in either cell lines or zebrafish?

More information about where ErbB kinases are expressed in human/mouse/zebrafish tissues would be informative. Is this a neutrophil intrinsic effect? The in vitro analysis supports this idea however it is unclear if this could be due to toxic effects of the compounds on neutrophils versus inhibition of a specific pathway.

Cell toxicity remains a concern (in particular Figure 4). Was there specificity of the inhibitors for ErbB kinases in neutrophils? Can a kinase activity assay be performed? What was the expression of ErbB kinases in human neutrophils? Was the activity/expression regulated by pro-survival signals?

Throughout the number of replicates and total N was not always clear. This should be indicated in the figures and figure legends.

Figure 1: The kinome figure is not user friendly. In Figure 1B, it would be useful to know what fold increase or decrease was observed with GMCSF treatment (and more information about the conditions of the treatment). Figure 1 should reference the tables that correspond with this data. An alternative would be to include a table within the figure (with fold changes in expression by GMCSF).

Figure 2: The kinetics of the assay was long duration (24 hour pre-treatment). What was the effect if larvae were treated with the drug for shorter times? What happens at earlier time points?

Figure 4: Is there a change in ErbB kinase expression or activity in normal/COPD donors? A western blot would be useful.

Figures 5 and 6: Do pro-survival signals increase ErbB kinase expression or activity?

Figure 7C: It was not entirely clear what is being measured here?

Figure 8: Was there an effect on neutrophil reverse migration or just apoptosis? The authors have reported the use of photoconversion to analyze neutrophil reverse migration and this would be informative in this context.

Reviewer #2:

This manuscript is from an established group of respected neutrophil biologists and the study combines profiling of the neutrophil kinome with studies of selected kinase inhibitors as a strategy to advance understanding of neutrophil apoptosis regulation and to identify possible targets for therapeutic intervention in COPD and related diseases.

The main strengths of the study are the generation of a kinome profile of human neutrophils and evidence of significant overlap with the kinome of zebrafish.

The main weaknesses are the lack of validation of the kinome data and the results of the inhibitor screen. Thus, the kinome analsysis remains merely a list of enzymes, and the results of the associated inhibitor screen have not been directly validated.

Essential revisions:

1) Rather than characterizing the kinome itself, a library of kinase inhibitors was screened to identify compounds that accelerate neutrophil apoptosis. Based on these results, further experiments focused on inhibitors of the receptor tyrosine kinase ErbB. However, these data are incomplete. Thus, as presented, the manuscript contains two incomplete stories.

2) Regarding the screen, what accounts for the differential effects of inhibitors that target the same kinases? Why does the ErbB1/B4 inhibitor induces more apoptosis than the ErbB1/B/B4 inhibitor (Figure 3)?

3) Most experiments focus on ErbB2 as an inhibitor target of interest, yet the kinome suggests that only ErbB3 is present. What accounts for this discrepancy? Data demonstrating the abundance of all ErbB family proteins in neutrophils should be added.

4) There is no direct measurement of ErbB isoform activity or phosphorylation or analysis of relevant substrates (in the absence or presence of inhibitors). Thus, the effects could be indirect or non-specific. To address this, the activity and abundance of ErbB family members in resting, apoptotic, and cAMP- or GMCSF-stimulated cells is missing. Reliance on inhibitors without demonstration of specificity and efficacy is risky.

Reviewer #3:

Rahman et al. use a drug discovery approach, combining an in vivo screen in zebrafish and an in vitro screen of human cells, to identify kinase inhibitors that would help resolve neutrophil-mediated inflammatory diseases. They focus on ErbB inhibitors and show that they promote neutrophil apoptosis, even in the presence of pro-survival stimuli. Then, they perform experiments in mouse and again in zebrafish to establish a proof-of-concept of the usefulness of this type of drugs in inflammatory diseases.

This combination of models makes for a powerful and exciting approach. In general, experiments are performed carefully, and the statistical analysis is sound. However, I find that the authors tend to overstate the novelty of their findings.

Inhibition of neutrophil function par erbstatin is not exactly something new – it has been described first in a 1990 paper (Naccache et al., 1990). Actually, searching PubMed with keywords "neutrophil" and "erbstatin" returns 38 references, of which not one is cited in the manuscript. Not all these references are relevant, but clearly the sentence at the end of the Introduction ("This study is the first to identifiy a role for ErbBs in neutrophil function and inflammation") is an exaggeration. The relevant previous literature has to be incorporated in Introduction and Discussion section.

In the Introduction, the authors state "there are no treatment strategies in clinical use to reverse this cellular mechanism": maybe not "reverse", but roflumilast is an approved anti-inflammatory drug to prevent exacerbation in COPD. This should be discussed at the very least.

The final zebrafish experiment (Figure 8) is a disappointing, given the easy imaging of neutrophil migration and death in this system. It does not add much to the initial screen. A measurement of frequency of neutrophil death at the site of injury, in particular, would be desirable considering the general emphasis of the manuscript on induction of neutrophil apoptosis.

Table S1A: Please provide a quantitation of the expression level of each kinase in human neutrophils. Please also provide the complete list and quantitation of kinases expressed in zebrafish neutrophils (as S1B, thus moving the common set as S1C). For both human and zebrafish genes, provide unique identifiers (GenBank or Ensembl IDs), not just names, which are often ambiguous.

[Editors’ note: what now follows is the decision letter after the authors submitted for further consideration.]

Thank you for resubmitting your work entitled "Inhibition of ErbB kinase signalling promotes resolution of neutrophilic inflammation" for further consideration at *eLife*. Your revised article has been favorably evaluated by Tadatsugu Taniguchi (Senior Editor), a Reviewing Editor, and three reviewers.

We feel that the manuscript has greatly been improved and clarified since the last submission. We were fortunate to have two of the original reviewers and a new reviewer, as the previous third was unavailable at this time. We especially appreciate the new reviewer for comments, as they often need to understand the assessments from the previous iteration, which is no longer present. Through the post-review discussions, we felt it unnecessary to respond to the main points of reviewer #3 but they may be helpful in your final edits. We mainly would like you to address the statistical analysis and points on CRISPR controls raised by reviewer #2. The full reviewer comments are below:

Reviewer #1:

This study is a very interesting, well-designed, and important advance to the field of neutrophil biology and apoptosis regulation. The use of 3 complementary models is a key strength. The data are thorough and convincing, and all points noted in prior critiques have been adequately addressed. No further revisions requested.

Kinome data were leveraged to identify and characterize ErbB family kinases as key regulators of neutrophil longevity. These data were extended to shown that ErbB pathway inhibitors can overcome pro-survival signaling to induce apoptosis and thereby reverse pathological neutrophil accumulation in disease states, including COPD.

Reviewer #2:

The authors report that inhibition of ErbB kinase limits neutrophil inflammation by affecting neutrophil apoptosis. The paper uses several systems: human neutrophils, zebrafish and mice to probe the effect of ErbB kinase inhibition on inflammation. The observations are intriguing and suggest that ErbB inhibition may provide a pathway with therapeutic importance for inflammatory disease. The revised manuscript is significantly improved and addressed the concerns raised in the prior review. I recommend the revised manuscript for publication in *eLife* after these concerns are addressed.

The controls for the crispants are not clearly outlined. How were the crispants validated? How many targets were used? In general, more than one target should be used for each gene.

Statistical analysis should be clarified in the legends and text.

Reviewer #3:

This study by Rahman et al. investigated the role of ErbB kinase in neutrophil apoptosis, using multiple approaches in vitro and in vivo. They observed that: (1) ErbB family were the common targets of compounds that leading to neutrophil apoptosis in both zebrafish inflammation model and human neutrophils; (2) the ErbBs inhibitors promoted the apoptosis of neutrophils from both healthy volunteers and COPD patients; (3) GMCSG and dbcAMP, the neutrophil survival stimuli, promoted ErbB2 and ErbB3 expression in human neutrophils; (4) tyrphostin AG825 (an ErbBs inhibitor) and knockdown of *egfra* and *erbb2* reduced inflammation in vivo. Altogether, the authors conclude that ErbB family participate in neutrophil survival and ErbB inhibitors play positive roles in accelerating inflammation resolution.

Overall, this study appears interesting, and data presented in this manuscript look solid. However, some data are confusing and do not fully support the conclusion of this study.

There are several issues that need to be addressed by the authors. Specific issues are referenced below.

Essential revisions:

1) Subsection “Identifying kinases regulating the resolution of neutrophilic inflammation in vivo*”*: "We quantified the ability of PKIS to reduce the number of neutrophils at the site of injury during the resolution phase of inflammation." Please provide the specific data. And how to determine the zebrafish inflammation model is in the phase of inflammatory resolution, not acute phase.

2) The authors just showed the statistic results about apoptosis assessed by flow cytometry. It would be better to show the specific flow images in different groups and compare the differences.

3) Detection apoptosis using light microscopy is not an ideal option. Maybe some apoptosis-related proteins should be detected by Western Blot, such as Bcl-2 and Bax.

4) In Figure 1B what does the x-axis represent?

5) The authors demonstrated that tyrphostin AG825-driven neutrophil apoptosis is caspase-dependent. It would be better to detect the caspase-3 expression by Western Blot.

6) The author showed that the expressions of both total and phosphorylated ErbB2 (pErbB2) were elevated. Does the increase of total protein indirectly increase the amount of phosphorylated protein? It would be better to detect the expression of phosphorylated ErbB2 by Western Blot. And the ratio pErbB2/ErbB2 should be shown.

7) The data in Figure 4 are not enough to support the conclusion. The flow cytometry images in different groups should be provided. Other methods for detecting neutrophil apoptosis in lung tissue should probably be provided, such as immunohistochemistry. Did Tyr accelerate the inflammatory resolution in lung tissue in zymosan treated mice? HE staining for lung tissue should be detected.

8) The authors didn't demonstrate the role of ErbB inhibitors in accelerating inflammatory resolution both in mice and zebrafish models. The process of inflammatory resolution is dynamic. It would be better to monitor the tissue injury dynamically.

---

## [Author Response]

[Editors’ note: the author responses to the first round of peer review follow.]

Within our discussion, the reviewers all felt that while the story on the ErbB2 role in inflammation was interesting, more work was needed to really conclude its specific role, especially as this is not the first study to identify a role for ErbBs in neutrophil function and inflammation. Moreover, there was a consensus that the manuscript, as it stands now, really consists of two incomplete and disconnected stories: the kinome and the inhibitor screen. They were also more puzzled that ErB2, which was the main focus of the paper, did not even come up in the kinome. This brings up questions about both approaches. Overall, the impression was that this paper would be better understood as two separate stories but that both stories needed more experiments to be convincing, as outlined in the specific comments, below. One of the reviewers points out that there is a conditional (floxed) Erb2 mutant in mouse (Ozcelik et al., 2002). If the authors use this line and a neutrophil-specific Cre line, they could add genetic evidence. They also agreed that kinase assays needed to be performed to show that the effects were specific as well as live imaging of their zebrafish experiments to determine whether the effects were on cell migration or cell death. Because we all felt that it was unlikely for you produce more convincing data in less than 2 months, the usual turnaround for eLife, any resubmission would need to be as a new paper, and we would argue without the kinome data.

On reflection, we agree that the description of the neutrophil kinome impaired the clarity of the subsequent experiments. We have therefore removed the kinome data (Figure 1). Comments relating to expression of ErbBs have resulted in new data in the form of a cellular ELISA and confirm ErbB3 protein expression by human neutrophils. We agree that adding genetic evidence would further support our conclusion. As a reviewer states, there is a conditional ErbB2 mutant in the mouse. However, we feel that a whole animal knockout may in fact be more informative since the use of ErbB inhibitors in an inflammatory disease setting, which is the long term aim of this work, would target multiple tissues and systems and not a single cell type. Both EGFR and ErbB2 whole organism knockouts in mice are limited by pre- and postnatal lethality. We have therefore addressed this in our zebrafish model of inflammation using CRISPR/Cas9 which recapitulate findings from the inhibitor studies. Furthermore, we show that targeting ErbBs induces neutrophil apoptosis in vivo, thus providing a potential mechanism for inflammation resolution, and this new data has been added to the manuscript as described in more detail below.

Reviewer #1:The authors report that inhibition of ErbB kinase limits neutrophil inflammation by affecting neutrophil apoptosis. The paper uses several systems: human neutrophils, zebrafish and mice to probe the effect of ErbB kinase inhibition on inflammation. The observations are intriguing and suggest that ErbB inhibition may provide a therapeutic target for inflammatory disease.

We thank the reviewer for their positive comments and are encouraged that they found our study interesting. We are glad that they were persuaded of the potential of ErbB inhibition as a therapeutic target for inflammatory diseases such as COPD.

Essential revisions:Genetic approaches to implicate ErbB kinases in these processes, especially in the zebrafish model or cell lines would better support their conclusion that ErbB kinases regulate neutrophil apoptosis and inflammation. Are there genetic mutants or knockdowns in either cell lines or zebrafish?

We agree with this reviewer that genetic approaches would provide better evidence for our conclusion that ErbB kinase inhibition increases apoptosis and enhances inflammation resolution. In order to directly address this, we have performed genetic knockdown in our zebrafish model using CRISPR/Cas9. This work forms new panels of novel data. We find that in zebrafish larvae with knockdown of both *egfra (EGFR/ERBB1* orthologue) and *erbb2* genes, there is a reduction in neutrophil number at the tail fin injury site (new Figure 5D), which recapitulates the phenotype seen in zebrafish larvae treated with ErbB inhibitors (new Figure 5B-C). Whole body neutrophil number was also reduced in these larvae (new Figures 5F and 5J).

More information about where ErbB kinases are expressed in human/mouse/zebrafish tissues would be informative. Is this a neutrophil intrinsic effect? The in vitro analysis supports this idea however it is unclear if this could be due to toxic effects of the compounds on neutrophils versus inhibition of a specific pathway.

EGFR is expressed in primary human neutrophils as confirmed by flow cytometry and CELISA (Lewkowicz et al., 2005) which is now referenced in the Discussion section. Datasets published on the online Expression Atlas (www.ebi.ac.uk) also confirm expression of ErbB family members in human and mouse neutrophils. Furthermore, we show human neutrophil expression of ErbB2 by western blot and RT-PCR (Figure 4) and have generated new data to show ErbB3 expression by ELISA (Figure 4C). Phosphorylated ErbB2 was detected in the Kinexus kinase antibody microarray (Table 1). We agree that the in vitro work suggests the effects of the inhibitors are mediated, at least in part, via intrinsic effects in neutrophils. Regarding toxic effects of inhibitors, we address this below.

Cell toxicity remains a concern (in particular Figure 4). Was there specificity of the inhibitors for ErbB kinases in neutrophils? Can a kinase activity assay be performed? What was the expression of ErbB kinases in human neutrophils? Was the activity/expression regulated by pro-survival signals?

We thank the reviewer for raising this point. We show in preliminary experiments that a 60KDa version of ErbB2 is upregulated by the survival factors GMCSF and dbcAMP (Figure 4). Unfortunately, kinase activity kits for ErbBs are not commercially available, but we have demonstrated that ErbB2 is rapidly phosphorylated in the presence of dbcAMP (Table 1). In new experimental work, we also show ErbB3 protein expression in human neutrophils by ELISA (new Figure 4C). Expression of ErbB3 did not change with the addition of survival factors dbcAMP, GMCSF or LPS. This assay however, measured total ErbB3 rather than phosphorylated ErbB3 and may suggest post-translational regulation is more important than protein levels. This would require further study and while very interesting, characterising ErbB expression is not the primary focus of this current manuscript. With respect to specificity, the inhibitors we have used are well-characterised, demonstrate selectivity for individual ErbB family members (see IC_50_ values below) and have been extensively used by others to study ErbBs. To add to strength to out conclusion, we have tested clinical ErbB inhibitors for their effect on neutrophil apoptosis (new Figure 2A). We agree however, that this does not preclude the possibility that they may have some off-target effects, and investigating this will form part of further studies in our laboratory. We argue that, since multiple ErbB-targeting compounds share a profound anti-inflammatory effect across several models and species, there is a strong likelihood of ErbBs being key to this process. Because human neutrophils are genetically intractable, we are unable to delete ErbB genes in these cells. However, new genetic evidence in zebrafish (as described above) further supports the conclusion that inhibition of ErbB signalling has anti-inflammatory effects.

IC_50_ values for selected inhibitors used in this study:

Gefitinib inhibits purified EGFR and HER-2 at IC_50_ values of 0.033 and ≥3.7 μmol/L, respectively (Wakeling et al., 2002)

Tyrphostin AG825 inhibits EGFR and ErbB2 at IC_50_ values of 19 and 0.35 μmol/L respectively (Osherov et al., 1993)

CP-724,714 inhibits EGFR and ErbB2 at IC_50_ values of 6.4 and 0.010 μmol/L respectively (Jani et al., 2007)

Erbstatin inhibits purified EGFR at a value of 0.77μmol/L (Umezawa et al., 1992)

Throughout the number of replicates and total N was not always clear. This should be indicated in the figures and figure legends.

We thank the reviewer for this comment and apologise for a lack of clarity. We have addressed this throughout the revised manuscript.

Figure 1: The kinome figure is not user friendly. In Figure 1B, it would be useful to know what fold increase or decrease was observed with GMCSF treatment (and more information about the conditions of the treatment). Figure 1 should reference the tables that correspond with this data. An alternative would be to include a table within the figure (with fold changes in expression by GMCSF).

This section has been removed from our manuscript to aid the overall clarity.

Figure 2: The kinetics of the assay was long duration (24 hour pre-treatment). What was the effect if larvae were treated with the drug for shorter times? What happens at earlier time points?

We apologise for this ambiguity. Figure 2 which detailed the zebrafish screen is now Figure 1A. There is no 24 hour pre-treatment in these experiments. The larvae are treated for a total of 6 hours, beginning at 6 hours after injury until 12 hours after injury. This timing is in order to capture those compounds which enhance inflammation resolution. We have added further detail to Figure 1—figure supplement 1 to clarify the timescale.

Figure 4: Is there a change in ErbB kinase expression or activity in normal/COPD donors? A western blot would be useful.

This is a very interesting point. Since the rates of either baseline or modified apoptosis did not differ between healthy controls and COPD patients, we hypothesised that ErbB expression also would not differ between these groups and therefore did not prioritise this experiment. Unfortunately, the ethical approvals for this study have now expired and we are unable to explore this further.

Figures 5 and 6: Do pro-survival signals increase ErbB kinase expression or activity?

The Kinexus antibody array (Table 1) shows upregulation of phosphorylated ErbB2 by dbcAMP at both 30 and 60 minutes. We also show both dbcAMP and GMCSF upregulate ErbB2 protein (Figure 4). Total ErbB3 expression was unaltered with dbcAMP, GMCSF or LPS (Figure 4). Surprisingly there are no well-optimised commercially available activity kits for ErbBs and since neutrophils are not transfectable we are unable to use plasmid-based reporter systems.

Figure 7C: It was not entirely clear what is being measured here?

We apologise for a lack of clarity. Figure 7C (now Figure 5C) shows the percentage neutrophil apoptosis on BALF cytocentrifuge slides from mice treated either with PBS (control) or tyrphostin (Tyr). The closed circles quantify ‘free’ (extracellular) neutrophils, i.e. not including neutrophils that have been ingested by macrophages. The closed triangles quantify both ‘free’ apoptotic neutrophils as well as those that are inside macrophages. This latter count takes into account apoptotic events (inclusions) that have been efferocytosed. Such apoptotic inclusions are highlighted by arrows in Figure 5E. We felt this was important to capture since ‘free’ apoptotic events in vivoare rare due to rapid efferocytosis. We have now made amendments to the figure legend to clarify this.

Figure 8: Was there an effect on neutrophil reverse migration or just apoptosis? The authors have reported the use of photoconversion to analyze neutrophil reverse migration and this would be informative in this context.

Figure 8 has been updated and is now a much expanded Figure 5. In addition to our original inhibitor data, we have added genetic evidence of a role of ErbB kinases in inflammation. We have also measured apoptosis in both inhibitor treated and *egfra* and *erbb2* ‘crispant’ larvae. Briefly, we find that neutrophil apoptosis was increased both at the tail fin injury site and in the caudal haematopoietic tissue of zebrafish treated overnight with ErbB inhibitors. Since we know from experience (https://dmm.biologists.org/content/dmm/9/6/621.full.pdf) that reverse migration is unlikely to occur in parallel with apoptosis, we did not prioritise photoconversion experiments. Apoptosis was not significantly increased in *egfra/erbb2* crispant larvae at the tail fin injury site, however potential explanations for this have been explored in the discussion section.

Reviewer #2:This manuscript is from an established group of respected neutrophil biologists and the study combines profiling of the neutrophil kinome with studies of selected kinase inhibitors as a strategy to advance understanding of neutrophil apoptosis regulation and to identify possible targets for therapeutic intervention in COPD and related diseases.The main strengths of the study are the generation of a kinome profile of human neutrophils and evidence of significant overlap with the kinome of zebrafish.The main weaknesses are the lack of validation of the kinome data and the results of the inhibitor screen. Thus, the kinome analsysis remains merely a list of enzymes, and the results of the associated inhibitor screen have not been directly validated.

We thank the reviewer for their positive comments. The weaknesses that they identified are addressed point by point below.

Essential revisions:1) Rather than characterizing the kinome itself, a library of kinase inhibitors was screened to identify compounds that accelerate neutrophil apoptosis. Based on these results, further experiments focused on inhibitors of the receptor tyrosine kinase ErbB. However, these data are incomplete. Thus, as presented, the manuscript contains two incomplete stories.

We agree with this reviewer that our original presentation of this work lacked clarity of the message we are trying to convey. In response to these comments, we have substantially rewritten the manuscript, removing the description of the kinome and focusing on the functional aspects by adding genetic evidence for the role of ErbB kinase in inflammation.

2) Regarding the screen, what accounts for the differential effects of inhibitors that target the same kinases? Why does the ErbB1/B4 inhibitor induces more apoptosis than the ErbB1/B/B4 inhibitor (Figure 3)?

These are interesting and insightful questions. Some of the effects are due to efficacy at the dose used, some due to cellular penetration of the drug (which we have not assessed) and other effects may relate to differential efficacy at the different receptors. Small changes in the balance of inhibition of one kinase over another might alter the final physiological effect. In addition, neutrophils are heterogenous cell populations with donor variability, so we would expect some variation in responses to treatment, even with the same compound.

3) Most experiments focus on ErbB2 as an inhibitor target of interest, yet the kinome suggests that only ErbB3 is present. What accounts for this discrepancy? Data demonstrating the abundance of all ErbB family proteins in neutrophils should be added.

We agree that our original manuscript did overly focus on ErbB2. This was primarily because the Kinexus antibody array (Table 1) identified only phosphorylated ErbB2, but indeed our inhibitor and genetic studies show roles for other ErbBs. We have now reduced the emphasis on ErbB2 in order to include reflect the potential of other ErbB family members as important targets. The kinome data used an arbitrary threshold and ErbB2 was present in those data at lower levels – these data have now been removed.

4) There is no direct measurement of ErbB isoform activity or phosphorylation or analysis of relevant substrates (in the absence or presence of inhibitors). Thus, the effects could be indirect or non-specific. To address this, the activity and abundance of ErbB family members in resting, apoptotic, and cAMP- or GMCSF-stimulated cells is missing. Reliance on inhibitors without demonstration of specificity and efficacy is risky.

We agree that we cannot rule out the possibility that some of the effects of the inhibitors may be exerted through other targets. This is difficult to circumvent, given that neutrophils are genetically intractable and therefore preclude the use of plasmid-based reporter systems. We have however, directly analysed the ErbB substrate AKT, and show that phosphorylation of AKT is reduced in the presence of tyrphostin (Figure 3F). Surprisingly and frustratingly there are no well-optimised commercially-available activity kits for ErbBs. With respect to abundance and activity, we show in the Kinexus antibody array (Table 1) that phosphorylated ErbB2 is upregulated by dbcAMP at both 30 and 60 minutes (ErbB2 is itself an ErbB substrate). We also show both dbcAMP and GMCSF upregulate ErbB2 protein (Figure 4). In new experimental work, we found that ErbB3 expression in human neutrophils was not regulated by dbcAMP, GMCSF or LPS (Figure 4, new panel C). This assay however, measured total ErbB3 rather than phosphorylated ErbB3 and may suggest post-translational regulation is more important than protein levels. Proving this would require further study and while very interesting, characterising ErbB expression is not the primary focus of this current manuscript. We argue that, since multiple structurally-unrelated ErbB-targeting compounds share a profound anti-inflammatory effect across several models and species, there is a strong likelihood of ErbBs being key to this process. This is supported by the new genetic knockdown of *egfra* and *erbb2* in zebrafish, which also reduces neutrophilic inflammation.

Reviewer #3:Rahman et al. use a drug discovery approach, combining an in vivo screen in zebrafish and an in vitro screen of human cells, to identify kinase inhibitors that would help resolve neutrophil-mediated inflammatory diseases. They focus on ErbB inhibitors and show that they promote neutrophil apoptosis, even in the presence of pro-survival stimuli. Then, they perform experiments in mouse and again in zebrafish to establish a proof-of-concept of the usefulness of this type of drugs in inflammatory diseases.This combination of models makes for a powerful and exciting approach. In general, experiments are performed carefully, and the statistical analysis is sound. However, I find that the authors tend to overstate the novelty of their findings.

We thank the reviewer for their positive comments and for recognising the powerful approaches we have taken and the care with which we have performed this work. We have now addressed the issue of novelty in our revision of the manuscript, please see below.

Inhibition of neutrophil function par erbstatin is not exactly something new – it has been described first in a 1990 paper (Naccache et al., 1990). Actually, searching PubMed with keywords "neutrophil" and "erbstatin" returns 38 references, of which not one is cited in the manuscript. Not all these references are relevant, but clearly the sentence at the end of the Introduction ("This study is the first to identify a role for ErbBs in neutrophil function and inflammation") is an exaggeration. The relevant previous literature has to be incorporated in Introduction and Discussion section.

We thank the reviewer for bringing this to our attention. We also agree that not all of the “neutrophil” and “erbstatin” references are relevant or appropriate, particularly since many of them have not directly measured effects of the inhibitors on neutrophil function. We have however, added studies to show an impact of erbstatin on ROS generation, chemotaxis and cellular signalling (listed below) (Discussion section). We have also added additional citations describing the use of tyrphostin inhibitors (not AG825 specifically) in in vivo models to the discussion. Furthermore, we have amended the final sentence of the Introduction to now state “This study reveals an opportunity for the use of ErbB inhibitors as a treatment for chronic neutrophilic inflammatory disease” to reflect the existing studies using erbstatin.

We have added the following citations:

Bierman et al., 2008

Takezawa et al., 2016

Shimizu et al., 2018

Dreiem et al., 2003

Mocsai et al., 1997

Yasui et al., 1994

al-Shami et al., 1997

In the Introduction, the authors state "there are no treatment strategies in clinical use to reverse this cellular mechanism": maybe not "reverse", but roflumilast is an approved anti-inflammatory drug to prevent exacerbation in COPD. This should be discussed at the very least.

We have now added a sentence and citations to discuss the use of roflumilast in COPD (see below). The sentence follows directly on from points referring specifically to targeting neutrophil cell death as a therapeutic strategy and we feel the statement is correct as written.

We have added the following citations to the Introduction:

Rabe et al., 2018

Martinez et al., 2018

The final zebrafish experiment (Figure 8) is a disappointing, given the easy imaging of neutrophil migration and death in this system. It does not add much to the initial screen. A measurement of frequency of neutrophil death at the site of injury, in particular, would be desirable considering the general emphasis of the manuscript on induction of neutrophil apoptosis.

This is a very important point and we have specifically addressed this. We

performed apoptosis studies in our zebrafish model which show that treatment of zebrafish larvae with ErbB inhibitors results in an increase in neutrophil apoptosis both at a site of injury (Figure 5H) and within the caudal haematopoietic tissue (Figure 5I). This suggests that ErbB inhibitors are able to induce neutrophil apoptosis both within a homeostatic environment, and at a site of inflammation in vivo. Neutrophil apoptosis at the tail fin injury site of zebrafish larvae with genetic knockdown of *egfra* and *erbb2* was also assessed (Figure 5J), however no significant differences were observed, as discussed previously.

Table S1A: Please provide a quantitation of the expression level of each kinase in human neutrophils. Please also provide the complete list and quantitation of kinases expressed in zebrafish neutrophils (as S1B, thus moving the common set as S1C). For both human and zebrafish genes, provide unique identifiers (GenBank or Ensembl IDs), not just names, which are often ambiguous.

As mentioned above, the description of the neutrophil kinome has been removed from our revised manuscript.

[Editors' note: the author responses to the re-review follow.]

We feel that the manuscript has greatly been improved and clarified since the last submission. We were fortunate to have two of the original reviewers and a new reviewer, as the previous third was unavailable at this time. We especially appreciate the new reviewer for comments, as they often need to understand the assessments from the previous iteration, which is no longer present. Through the post-review discussions, we felt it unnecessary to respond to the main points of reviewer #3 but they may be helpful in your final edits. We mainly would like you to address the statistical analysis and points on CRISPR controls raised by reviewer #2.

On behalf of all authors, we are extremely grateful for further evaluation of our manuscript, and for noting the improvements we have made. We also greatly appreciate the continuity of reviewers (as well as reviewer #3 who has taken the time to evaluate our paper), and for giving us the opportunity to respond to these comments with a revised manuscript. We take note that post-review discussions require us to mainly address the statistical analysis and points on CRISPR controls.

The full reviewer comments are below:Reviewer #1:This study is a very interesting, well-designed, and important advance to the field of neutrophil biology and apoptosis regulation. The use of 3 complementary models is a key strength. The data are thorough and convincing, and all points noted in prior critiques have been adequately addressed. No further revisions requested.Kinome data were leveraged to identify and characterize ErbB family kinases as key regulators of neutrophil longevity. These data were extended to shown that ErbB pathway inhibitors can overcome pro-survival signaling to induce apoptosis and thereby reverse pathological neutrophil accumulation in disease states, including COPD.

We thank reviewer #1 for their time and expertise in evaluating our manuscript for a second time. We are very encouraged by these positive comments.

Reviewer #2:The authors report that inhibition of ErbB kinase limits neutrophil inflammation by affecting neutrophil apoptosis. The paper uses several systems: human neutrophils, zebrafish and mice to probe the effect of ErbB kinase inhibition on inflammation. The observations are intriguing and suggest that ErbB inhibition may provide a pathway with therapeutic importance for inflammatory disease. The revised manuscript is significantly improved and addressed the concerns raised in the prior review. I recommend the revised manuscript for publication in eLife after these concerns are addressed.

We sincerely thank reviewer #2 for this second critique of our manuscript and are encouraged by the positive review. We are delighted to be able to respond to the comments in a revised manuscript.

The controls for the crispants are not clearly outlined. How were the crispants validated? How many targets were used? In general, more than one target should be used for each gene.

We apologise for this ambiguity. We have added the following to subsection “Generation of transient CRISPR/Cas9 zebrafish mutants”: “The non-targeting control in these experiments was a guide RNA targeted towards tyrosinase, a gene involved in pigment formation and therefore easy to identify when mutated, and which is used by others in the field as a CRISPR/Cas9 control (O'Connor et al., 2019; Varshney et al., 2016). We have previously shown that this guide does not influence neutrophilic inflammation in the zebrafish (Evans et al., 2019; Isles et al., 2019).” Gene mutation in crispants was validated by melt curve analysis. We have expanded on this in subsection “Genotyping of crispant larvae” by quoting the average mutation rate. We also determined that these larvae were generally healthy by observing a normal swim bladder at 5 dpf – a sensitive assay for generalised developmental defects. We have used one guide RNA per gene. Since we demonstrated an efficient mutation rate as well as a clear phenotype, we did not feel compelled to use additional guides.

Statistical analysis should be clarified in the legends and text.

Thank you for highlighting this important omission. We have now added full details of the statistical analysis to each figure legend and refer the reader to legends in subsection “Statistical Analysis”.

Reviewer #3:This study by Rahman et al. investigated the role of ErbB kinase in neutrophil apoptosis, using multiple approaches in vitro and in vivo. They observed that: (1) ErbB family were the common targets of compounds that leading to neutrophil apoptosis in both zebrafish inflammation model and human neutrophils; (2) the ErbBs inhibitors promoted the apoptosis of neutrophils from both healthy volunteers and COPD patients; (3) GMCSG and dbcAMP, the neutrophil survival stimuli, promoted ErbB2 and ErbB3 expression in human neutrophils; (4) tyrphostin AG825 (an ErbBs inhibitor) and knockdown of egfra and erbb2 reduced inflammation in vivo. Altogether, the authors conclude that ErbB family participate in neutrophil survival and ErbB inhibitors play positive roles in accelerating inflammation resolution.Overall, this study appears interesting, and data presented in this manuscript look solid. However, some data are confusing and do not fully support the conclusion of this study.There are several issues that need to be addressed by the authors. Specific issues are referenced below.

We are grateful to reviewer #3 for evaluating our manuscript, and for suggesting modifications for improvement. Although post-review discussions concluded that these were not all necessary to carry out, we have made revisions based on these suggestions (detailed below).

Essential revisions:1) Subsection “Identifying kinases regulating the resolution of neutrophilic inflammation in vivo”: "We quantified the ability of PKIS to reduce the number of neutrophils at the site of injury during the resolution phase of inflammation." Please provide the specific data. And how to determine the zebrafish inflammation model is in the phase of inflammatory resolution, not acute phase.

We have now added “recruitment phase” and “resolution phase” to Figure 1—figure supplement 1 as well as a citation which establishes the timecourse of neutrophilic inflammation in this model.

2) The authors just showed the statistic results about apoptosis assessed by flow cytometry. It would be better to show the specific flow images in different groups and compare the differences.

Thank you for this excellent suggestion. We have now added representative flow cytometry dot plots as a new figure supplement (Figure 1—figure supplement 2).

3) Detection apoptosis using light microscopy is not an ideal option. Maybe some apoptosis-related proteins should be detected by Western Blot, such as Bcl-2 and Bax.

Morphological assessment of neutrophil apoptosis is considered to be the gold standard. We have used a combination of light microscopy (a morphological assay) and flow cytometry (a biochemical assay) to assess neutrophil apoptosis in this paper, as well as showing the cell death was a caspase-dependent mechanism.

4) In Figure 1B what does the x-axis represent?

We apologise for any ambiguity. We have changed Figure 1B x-axis to “Individual protein kinase inhibitor compounds [62.5 μM]”.

5) The authors demonstrated that tyrphostin AG825-driven neutrophil apoptosis is caspase-dependent. It would be better to detect the caspase-3 expression by Western Blot.

As stated above, we feel that showing both Annexin V positivity as well as inhibition of cell death by Q-VD-OPh (Figure 2—figure supplement 1) are robust measurements of apoptosis.

6) The author showed that the expressions of both total and phosphorylated ErbB2 (pErbB2) were elevated. Does the increase of total protein indirectly increase the amount of phosphorylated protein? It would be better to detect the expression of phosphorylated ErbB2 by Western Blot. And the ratio pErbB2/ErbB2 should be shown.

We were unable to find a clean and reliable antibody to pErbB2 that worked well in neutrophil lysates by Western blot.

7) The data in Figure 4 are not enough to support the conclusion. The flow cytometry images in different groups should be provided. Other methods for detecting neutrophil apoptosis in lung tissue should probably be provided, such as immunohistochemistry. Did Tyr accelerate the inflammatory resolution in lung tissue in zymosan treated mice? HE staining for lung tissue should be detected.

We apologise for any confusion caused. The data in Figure 4A-E (now Figure 5) data from light microscopy and this is stated in the legend. Other methods for detecting neutrophil apoptosis in lung tissue should probably be provided, such as immunohistochemistry. Did Tyr accelerate the inflammatory resolution in lung tissue in zymosan treated mice? HE staining for lung tissue should be detected. Thank you for this suggestion. Neutrophil apoptosis was enumerated in BALF by light microscopy, which is considered the gold standard. Unfortunately, lungs were not obtained from the zymosan peritonitis animals.

8) The authors didn't demonstrate the role of ErbB inhibitors in accelerating inflammatory resolution both in mice and zebrafish models. The process of inflammatory resolution is dynamic. It would be better to monitor the tissue injury dynamically.

We thank the reviewer for their comment. Measuring tissue injury dynamically in the mouse is very challenging and beyond the scope of this study. Because we have studied multiple timepoints in the fish we feel this gives information on the dynamics of inflammation resolution.